# Quantum Algorithms for Sampling Log-Concave Distributions and Estimating Normalizing Constants

**Andrew M. Childs**

Joint Center for Quantum Information and Computer Science,
Department of Computer Science, and
Institute for Advanced Computer Studies
University of Maryland
`amchilds@umd.edu`

**Tongyang Li**

Center on Frontiers of Computing Studies and
School of Computer Science
Peking University
`tongyangli@pku.edu.cn`

**Jin-Peng Liu**

Simons Institute and
Department of Mathematics
UC Berkeley
`jliu1219@terpmail.umd.edu`

**Chunhao Wang**

Department of Computer Science and Engineering
Pennsylvania State University
`cwang@psu.edu`

**Ruizhe Zhang**

Department of Computer Science
The University of Texas at Austin
`ruizhe@utexas.edu`

## Abstract

Given a convex function $f \colon \mathbb{R}^d \to \mathbb{R}$, the problem of sampling from a distribution $\propto e^{-f(x)}$ is called log-concave sampling. This task has wide applications in machine learning, physics, statistics, etc. In this work, we develop quantum algorithms for sampling log-concave distributions and for estimating their normalizing constants $\int_{\mathbb{R}^d} e^{-f(x)} \mathrm{d}x$. First, we use underdamped Langevin diffusion to develop quantum algorithms that match the query complexity (in terms of the condition number $\kappa$ and dimension $d$) of analogous classical algorithms that use gradient (first-order) queries, even though the quantum algorithms use only evaluation (zeroth-order) queries. For estimating normalizing constants, these algorithms also achieve quadratic speedup in the multiplicative error $\epsilon$. Second, we develop quantum Metropolis-adjusted Langevin algorithms with query complexity $\widetilde{O}(\kappa^{1/2}d)$ and $\widetilde{O}(\kappa^{1/2}d^{3/2}/\epsilon)$ for log-concave sampling and normalizing constant estimation, respectively, achieving polynomial speedups in $\kappa, d, \epsilon$ over the best known classical algorithms by exploiting quantum analogs of the Monte Carlo method and quantum walks. We also prove a $1/\epsilon^{1-o(1)}$ quantum lower bound for estimating normalizing constants, implying near-optimality of our quantum algorithms in $\epsilon$.

36th Conference on Neural Information Processing Systems (NeurIPS 2022).

# 1 Introduction

Sampling from a given distribution is a fundamental computational problem. For example, in statistics, samples can determine confidence intervals or explore posterior distributions. In machine learning, samples are used for regression and to train supervised learning models. In optimization, samples from well-chosen distributions can produce points near local or even global optima.

Sampling can be nontrivial even when the distribution is known. Indeed, efficient sampling is often a challenging computational problem, and bottlenecks the running time in many applications. Many efforts have been made to develop fast sampling methods. Among those, one of the most successful tools is Markov Chain Monte Carlo (MCMC), which uses a Markov chain that converges to the desired distribution to (approximately) sample from it.

Here we focus on the fundamental task of *log-concave sampling*, i.e., sampling from a distribution proportional to $e^{-f}$ where $f \colon \mathbb{R}^d \to \mathbb{R}$ is a convex function. This covers many practical applications such as multivariate Gaussian distributions and exponential distributions. Provable performance guarantees for log-concave sampling have been widely studied [15]. A closely related problem is estimating the normalizing constants of log-concave distributions, which also has many applications [16].

Quantum computing has been applied to speed up many classical algorithms based on Markov processes, so it is natural to investigate quantum algorithms for log-concave sampling. If we can prepare a quantum state whose amplitudes are the square roots of the corresponding probabilities, then measurement yields a random sample from the desired distribution. In this approach, the number of required qubits is only poly-logarithmic in the size of the sample space. Unfortunately, such a quantum state probably cannot be efficiently prepared in general, since this would imply $\mathsf{SZK} \subseteq \mathsf{BQP}$ [1]. Nevertheless, in some cases, quantum algorithms can achieve polynomial speedup over classical algorithms. Examples include uniform sampling on a 2D lattice [35], estimating partition functions [4, 22, 31, 45, 46], and estimating volumes of convex bodies [6]. However, despite the importance of sampling log-concave distributions and estimating normalizing constants, we are not aware of any previous quantum speedups for general instances of these problems.

**Formulation**   In this paper, we consider a $d$-dimensional convex function $f \colon \mathbb{R}^d \to \mathbb{R}$ which is $L$-smooth and $\mu$-strongly convex, i.e., $\mu, L > 0$ and for any $x, y \in \mathbb{R}^d$, $x \neq y$,

$$\frac{f(y) - f(x) - \langle \nabla f(x), y - x \rangle}{\|x - y\|_2^2 / 2} \in [\mu, L]. \tag{1.1}$$

We denote by $\kappa := L/\mu$ the condition number of $f$. The corresponding log-concave distribution has probability density $\rho_f \colon \mathbb{R}^d \to \mathbb{R}$ with

$$\rho_f(x) := \frac{e^{-f(x)}}{Z_f}, \tag{1.2}$$

where the normalizing constant is

$$Z_f := \int_{x \in \mathbb{R}^d} e^{-f(x)} \, \mathrm{d}x. \tag{1.3}$$

When there is no ambiguity, we abbreviate $\rho_f$ and $Z_f$ as $\rho$ and $Z$, respectively. Given an $\epsilon \in (0, 1)$,

- the goal of *log-concave sampling* is to output a random variable with distribution $\widetilde{\rho}$ such that $\|\widetilde{\rho} - \rho\| \leq \epsilon$, and

- the goal of *normalizing constant estimation* is to output a value $\widetilde{Z}$ such that with probability at least $2/3$, $(1 - \epsilon)Z \leq \widetilde{Z} \leq (1 + \epsilon)Z$.

Here $\| \cdot \|$ is a certain norm. We consider the general setting where the function $f$ is specified by an oracle. In particular, we consider the quantum evaluation oracle $O_f$, a standard model in the quantum computing literature [3, 6, 7, 50]. The evaluation oracle acts as

$$O_f|x, y\rangle = |x, f(x) + y\rangle \quad \forall x \in \mathbb{R}^d, y \in \mathbb{R}. \tag{1.4}$$

(Quantum computing notations are briefly explained in Section 2.) We also consider the quantum gradient oracle $O_{\nabla f}$ with

$$O_{\nabla f}|x, z\rangle = |x, \nabla f(x) + z\rangle \quad \forall x, z \in \mathbb{R}^d. \tag{1.5}$$

In other words, we allow superpositions of queries to both function evaluations and gradients. The essence of quantum speedup is the ability to compute with carefully designed superpositions.

**Contributions**  Our main results are quantum algorithms that speed up log-concave sampling and normalizing constant estimation.

**Theorem 1.1** (Main log-concave sampling result). *Let $\rho$ denote the log-concave distribution (1.2). There exist quantum algorithms that output a random variable distributed according to $\widetilde{\rho}$ such that*

- *$W_2(\widetilde{\rho}, \rho) \leq \epsilon$ where $W_2$ is the Wasserstein 2-norm (2.4), using $\widetilde{O}(\kappa^{7/6}d^{1/6}\epsilon^{-1/3} + \kappa d^{1/3}\epsilon^{-2/3})$ queries to the quantum evaluation oracle (1.4); or*
- *$\|\widetilde{\rho} - \rho\|_{TV} \leq \epsilon$ where $\|\cdot\|_{TV}$ is the total variation distance (2.3), using $\widetilde{O}(\kappa^{1/2}d)$ queries to the quantum gradient oracle (1.5), or $\widetilde{O}(\kappa^{1/2}d^{1/4})$ queries when the initial distribution is warm (formally defined in Appendix C.2.1).*

In the above results, the query complexity $\widetilde{O}(\kappa^{7/6}d^{1/6}\epsilon^{-1/3} + \kappa d^{1/3}\epsilon^{-2/3})$ is achieved by our quantum ULD-RMM algorithm. Although the quantum query complexity is the same as the best known classical result [37], we emphasize that our quantum algorithm uses a zeroth-order oracle while [37] uses a first-order oracle. The query complexity $\widetilde{O}(\kappa^{1/2}d)$ is achieved by our quantum MALA algorithm that uses a first-order oracle (as in classical algorithms). This is a quadratic speedup in $\kappa$ compared with the best known classical algorithm [28]. With a warm start, our quantum speedup is even more significant: we achieve quadratic speedups in $\kappa$ and $d$ as compared with the best known classical algorithm with a warm start [47].

**Theorem 1.2** (Main normalizing constant estimation result). *There exist quantum algorithms that estimate the normalizing constant by $\widetilde{Z}$ within multiplicative error $\epsilon$ with probability at least $3/4$,*

- *using $\widetilde{O}(\kappa^{7/6}d^{7/6}\epsilon^{-1} + \kappa d^{4/3}\epsilon^{-1})$ queries to the quantum evaluation oracle (1.4); or*
- *using $\widetilde{O}(\kappa^{1/2}d^{3/2}\epsilon^{-1})$ queries to the quantum gradient oracle (1.5).*

*Furthermore, this task has quantum query complexity at least $\Omega(\epsilon^{-1+o(1)})$ (Theorem 5.1).*

Our query complexity $\widetilde{O}(\kappa^{7/6}d^{7/6}\epsilon^{-1} + \kappa d^{4/3}\epsilon^{-1})$ for normalizing constant estimation achieves a quadratic speedup in precision compared with the best known classical algorithm [16]. More remarkably, our quantum ULD-RMM algorithm again uses a zeroth-order oracle while the slower best known classical algorithm uses a first-order oracle [16]. Our quantum algorithm working with a first-order oracle achieves polynomial speedups in all parameters compared with the best known classical algorithm [16]. Moreover, the precision-dependence of our quantum algorithms is nearly optimal, which is quadratically better than the classical lower bound in $1/\epsilon$ [16].

To the best of our knowledge, these are the first quantum algorithms with quantum speedup for the fundamental problems of log-concave sampling and estimating normalizing constants. We explore multiple classical techniques including the underdamped Langevin diffusion (ULD) method [12–14, 43], the randomized midpoint method for underdamped Langevin diffusion (ULD-RMM) [36, 37], and the Metropolis adjusted Langevin algorithm (MALA) [8, 11, 15, 28, 29, 47], and achieve quantum speedups. Our main contributions are as follows.

- *Log-concave sampling.*  For this problem, our quantum algorithms based on ULD and ULD-RMM have the same query complexity as the best known classical algorithms, but our quantum algorithms only use a zeroth-order (evaluation) oracle, while the classical algorithms use the first-order (gradient) oracle. For MALA, this improvement on the order of oracles is nontrivial, but we can use the quantum gradient oracle in our quantum MALA algorithm to achieve a quadratic speedup in the condition number $\kappa$. Furthermore, given a warm-start distribution, our quantum algorithm achieves a quadratic speedup in all parameters.

- *Normalizing constant estimation.*  For this problem, our quantum algorithms provide larger speedups. In particular, our quantum algorithms based on ULD and ULD-RMM achieve quadratic

speedup in the multiplicative precision $\epsilon$ (while using a zeroth-order oracle) compared with the corresponding best-known classical algorithms (using a first-order oracle). Our quantum algorithm based on MALA achieves polynomial speedups in all parameters. Furthermore, we prove that our quantum algorithm is nearly optimal in terms of $\epsilon$.

We summarize our results and compare them to previous classical algorithms in Table 1 and Table 2. See Appendix A for more detailed comparisons to related classical and quantum work.

Table 1: Summary of the query complexities of classical and quantum algorithms for sampling a $d$-dimensional log-concave distribution. Here $\kappa = L/\mu$ in (1.1) and $\epsilon$ is the error in the designated norm.

| Method | Oracle | Complexity | Norm |
|---|---|---|---|
| ULD [10] | gradient | $\widetilde{O}\left(\kappa^2 d^{1/2}\epsilon^{-1}\right)$ | $W_2$ |
| ULD-RMM [37] | gradient | $\widetilde{O}\left(\kappa^{7/6}d^{1/6}\epsilon^{-1/3} + \kappa d^{1/3}\epsilon^{-2/3}\right)$ | $W_2$ |
| MALA [28] | gradient | $\widetilde{O}(\kappa d)$ | TV |
| MALA with warm start [47] | gradient | $\widetilde{O}\left(\kappa d^{1/2}\right)$ | TV |
| Quantum Inexact ULD (Theorem C.1) | evaluation | $\widetilde{O}\left(\kappa^2 d^{1/2}\epsilon^{-1}\right)$ | $W_2$ |
| Quantum Inexact ULD-RMM (Theorem C.2) | evaluation | $\widetilde{O}\left(\kappa^{7/6}d^{1/6}\epsilon^{-1/3} + \kappa d^{1/3}\epsilon^{-2/3}\right)$ | $W_2$ |
| Quantum MALA (Theorem C.7) | gradient | $\widetilde{O}\left(\kappa^{1/2}d\right)$ | TV |
| Quantum MALA (warm start) (Theorem C.6) | gradient | $\widetilde{O}\left(\kappa^{1/2}d^{1/4}\right)$ | TV |

Table 2: Summary of the query complexities of classical and quantum algorithms for estimating the normalizing constant of a $d$-dimensional log-concave distribution. Here $\kappa = L/\mu$ in (1.1) and $\epsilon$ is the multiplicative error.

| Method | Oracle | Complexity |
|---|---|---|
| Multilevel ULD [16] | gradient | $\widetilde{O}\left(\kappa^2 d^{3/2}\epsilon^{-2}\right)$ |
| Multilevel ULD-RMM [16] | gradient | $\widetilde{O}\left(\kappa^{7/6}d^{7/6}\epsilon^{-2} + \kappa d^{4/3}\epsilon^{-2}\right)$ |
| MALA [16] | gradient | $\widetilde{O}\left(\kappa d^2\epsilon^{-2}\max\{1, \frac{\kappa}{d}\}\right)$ |
| Multilevel Quantum Inexact ULD (Theorem D.3) | evaluation | $\widetilde{O}\left(\kappa^2 d^{3/2}\epsilon^{-1}\right)$ |
| Multilevel Quantum Inexact ULD-RMM (Theorem D.4) | evaluation | $\widetilde{O}\left(\kappa^{7/6}d^{7/6}\epsilon^{-1} + \kappa d^{4/3}\epsilon^{-1}\right)$ |
| Quantum annealing with Quantum MALA (Theorem D.2) | gradient | $\widetilde{O}\left(\kappa^{1/2}d^{3/2}\epsilon^{-1}\right)$ |

**Techniques** In this work, we develop a systematic approach for studying the complexity of quantum walk mixing and show that for any reversible classical Markov chain, we can obtain quadratic speedup for the mixing time as long as the initial distribution is warm. In particular, we apply the quantum walk and quantum annealing in the context of Langevin dynamics and achieve polynomial quantum speedups.

The technical ingredients of our quantum algorithms are highlighted below.

- *Quantum simulated annealing (Lemma 3.2).* Our quantum algorithm for estimating normalizing constants combines the quantum simulated annealing framework of [45] and the quantum mean estimation algorithm of [31]. For each type of Langevin dynamics (which are random walks), we build a corresponding quantum walk. Crucially, the spectral gap of the random walk is quadratically amplified in the phase gap of the corresponding quantum walk. This allows us to use a Grover-like procedure to produce the stationary distribution state given a sufficiently good initial state. In the simulated annealing framework, this initial state is the stationary distribution state of the previous Markov chain.

- *Effective spectral gap (Lemma C.7).* We show how to leverage a "warm" initial distribution to achieve a quantum speedup for sampling. Classically, a warm start leads to faster mixing even if

the spectral gap is small. Quantumly, we generalize the notion of "effective spectral gap" [6, 27, 34] to our more general sampling problem. We show that with a bounded warmness parameter, quantum algorithms can achieve a quadratic speedup in the mixing time. By viewing the sampling problem as a simulated annealing process with only one Markov chain, we prove a quadratic speedup for quantum MALA by analyzing the effective spectral gap.

- *Quantum gradient estimation (Lemma C.1).* We adapt Jordan's quantum gradient algorithm [24] to the ULD and ULD-RMM algorithms and give rigorous proofs to bound the sampling error due to gradient estimation errors.

**Open questions**    Our work raises several natural questions for future investigation:

- Can we achieve quantum speedup in $d$ and $\kappa$ for unadjusted Langevin algorithms such as ULD and ULD-RMM? The main difficulty is that ULD and ULD-RMM are irreversible, while most available quantum walk techniques only apply to reversible Markov chains. New techniques might be required to resolve this question.
- Can we achieve further quantum speedup for estimating normalizing constants with a warm start distribution? This might require a more refined version of quantum mean estimation.
- Can we give quantum algorithms for estimating normalizing constants with query complexity sublinear in $d$? Such a result would give a provable quantum-classical separation due to the $\Omega(d^{1-o(1)}/\epsilon^{2-o(1)})$ classical lower bound proved in [16].

**Limitations and societal impacts**    Researchers working on theoretical aspects of quantum computing or Monte Carlo methods may benefit from our results. In the long term, once fault-tolerant quantum computers have been built, our results may find practical applications in MCMC methods arising in the real world. As far as we are aware, our work does not have negative societal impacts.

## 2   Preliminaries

**Basic definitions of quantum computation**    Quantum mechanics is formulated in terms of linear algebra. The *computational basis* of $\mathbb{C}^d$ is $\{\vec{e}_0, \ldots, \vec{e}_{d-1}\}$, where $\vec{e}_i = (0, \ldots, 1, \ldots, 0)^\top$ with the 1 in the $(i+1)^{\text{st}}$ position. We use *Dirac notation*, writing $|i\rangle$ (called a "ket") for $\vec{e}_i$ and $\langle i|$ (a "bra") for $\vec{e}_i^\top$.

The *tensor product* of quantum states is their Kronecker product: if $|u\rangle \in \mathbb{C}^{d_1}$ and $|v\rangle \in \mathbb{C}^{d_2}$, then we have $|u\rangle \otimes |v\rangle \in \mathbb{C}^{d_1} \otimes \mathbb{C}^{d_2}$ with

$$|u\rangle \otimes |v\rangle = (u_0 v_0, u_0 v_1, \ldots, u_{d_1-1} v_{d_2-1})^\top. \tag{2.1}$$

The basic element of quantum information is a *qubit*, a quantum state in $\mathbb{C}^2$, which can be written as $a|0\rangle + b|1\rangle$ for some $a, b \in \mathbb{C}$ with $|a|^2 + |b|^2 = 1$. An $n$-qubit tensor product state can be written as $|v_1\rangle \otimes \cdots \otimes |v_n\rangle \in \mathbb{C}^{2^n}$, where for any $i \in [n]$, $|v_i\rangle$ is a one-qubit state. Note however that most states in $\mathbb{C}^{2^n}$ are not product states. We sometimes abbreviate $|u\rangle \otimes |v\rangle$ as $|u\rangle |v\rangle$.

Operations on quantum states are *unitary transformations*. They are typically stated in the circuit model, where a $k$-*qubit gate* is a unitary matrix in $\mathbb{C}^{2^k}$. Two-qubit gates are *universal*, i.e., every $n$-qubit gate can be decomposed into a product of gates that act as the identity on $n-2$ qubits and as some two-qubit gate on the other 2 qubits. The *gate complexity* of an operation refers to the number of two-qubit gates used in a quantum circuit for realizing it.

Quantum access to a function, referred to as a *quantum oracle*, must be reversible and allow access to different values of the function in *superposition* (i.e., for linear combinations of computational basis states). For example, consider the unitary evaluation oracle $O_f$ defined in (1.4). Given a probability distribution $\{p_i\}_{i=1}^n$ and a set of points $\{x_i\}_{i=1}^n$, we have

$$O_f \sum_{i=1}^n \sqrt{p_i} |x_i\rangle |0\rangle = \sum_{i=1}^n \sqrt{p_i} |x_i\rangle |f(x_i)\rangle. \tag{2.2}$$

Then a measurement would give $f(x_i)$ with probability $p_i$. However, a quantum oracle can not only simulate random sampling, but can enable uniquely quantum behavior through interference. Examples include amplitude amplification—the main idea behind Grover's search algorithm [20] and

the amplitude estimation procedure used in this paper—and many other quantum algorithms relying on coherent quantum access to a function. Similar arguments apply to the quantum gradient oracle (1.5). If a classical oracle can be computed by an explicit classical circuit, then the corresponding quantum oracle can be implemented by a quantum circuit of approximately the same size. Therefore, these quantum oracles provide a useful framework for understanding the quantum complexity of log-concave sampling and normalizing constant estimation.

To sample from a distribution $\pi$, it suffices to prepare the state $|\pi\rangle := \sum_x \sqrt{\pi_x}|x\rangle$ and then measure it. For a Markov chain specified by a transition matrix $P$ with stationary distribution $\pi$, one can construct a corresponding *quantum walk* operator $W(P)$. Intuitively, quantum walks can be viewed as applying a sequence of quantum unitaries on a quantum state encoding the initial distribution to rotate it to the subspace of stationary distribution $|\pi\rangle$. The number of rotations needed (i.e., the angle between the initial distribution and stationary distribution) depends on the spectral gap of $P$, and a quantum algorithm can achieve a quadratic speedup via *quantum phase estimation* and *amplification* algorithms. More background on quantum walk is given in Appendix C.2.2.

**Notations**   Throughout the paper, the big-O notations $O(\cdot)$, $o(\cdot)$, $\Omega(\cdot)$, and $\Theta(\cdot)$ follow common definitions. The $\tilde{O}$ notation omits poly-logarithmic terms, i.e., $\tilde{O}(f) := O(f\,\mathrm{poly}(\log f))$. We say a function $f$ is *L-Lipschitz continuous* at $x$ if $|f(x) - f(y)| \leq L\|x - y\|$ for all $y$ sufficiently near $x$. The total variation distance (TV-distance) between two functions $f, g \colon \mathbb{R}^d \to \mathbb{R}$ is defined as

$$\|f - g\|_{\mathrm{TV}} := \frac{1}{2}\int_{\mathbb{R}^d}|f(x) - g(x)|\,\mathrm{d}x. \tag{2.3}$$

Let $\mathcal{B}(\mathbb{R}^d)$ denote the Borel $\sigma$-field of $\mathbb{R}^d$. Given probability measures $\mu$ and $\nu$ on $(\mathbb{R}^d, \mathcal{B}(\mathbb{R}^d))$, a *transference plan* $\zeta$ between $\mu$ and $\nu$ is defined as a probability measure on $(\mathbb{R}^d \times \mathbb{R}^d, \mathcal{B}(\mathbb{R}^d) \times \mathcal{B}(\mathbb{R}^d))$ such that for any $A \subseteq \mathbb{R}^d$, $\zeta(A \times \mathbb{R}^d) = \mu(A)$ and $\zeta(\mathbb{R}^d \times A) = \nu(A)$. We let $\Gamma(\mu, \nu)$ denote the set of all transference plans. We let

$$W_2(\mu, \nu) := \left(\inf_{\zeta \in \Gamma(\mu,\nu)} \int_{\mathbb{R}^d \times \mathbb{R}^d} \|x - y\|_2^2 \,\mathrm{d}\zeta(x,y)\right)^{\frac{1}{2}} \tag{2.4}$$

denote the Wasserstein 2-norm between $\mu$ and $\nu$.

# 3   Quantum Algorithm for Log-Concave Sampling

In this section, we describe several quantum algorithms for sampling log-concave distributions.

**Quantum inexact ULD and ULD-RMM**   We first show that the gradient oracle in the classical ULD and ULD-RMM algorithms can be efficiently simulated by the quantum evaluation oracle via quantum gradient estimation. Suppose we are given access to the evaluation oracle (1.4) for $f(x)$. Then by Jordan's algorithm [24] (see Lemma C.1 for details), there is a quantum algorithm that can compute $\nabla f(x)$ with a polynomially small $\ell_1$-error by querying the evaluation oracle $O(1)$ times. Using this, we can prove the following theorem (see Appendix C.1 for details).

**Theorem 3.1** (Informal version of Theorem C.1 and Theorem C.2). *Let $\rho \propto e^{-f}$ be a d-dimensional log-concave distribution with $f$ satisfying (1.1). Given a quantum evaluation oracle for $f$,*

- *the quantum inexact ULD algorithm uses $\tilde{O}(\kappa^2 d^{1/2}\epsilon^{-1})$ queries, and*
- *the quantum inexact ULD-RMM algorithm uses $\tilde{O}(\kappa^{7/6} d^{1/6}\epsilon^{-1/3} + \kappa d^{1/3}\epsilon^{-2/3})$ queries,*

*to quantumly sample from a distribution that is $\epsilon$-close to $\rho$ in $W_2$-distance.*

We note that the query complexities of our quantum algorithms using a *zeroth-order* oracle match the state-of-the-art classical ULD [10] and ULD-RMM [37] complexities with a *first-order* oracle. The main technical difficulty of applying the quantum gradient algorithm is that it produces a *stochastic gradient oracle* in which the output of the quantum algorithm $\mathbf{g}$ satisfies $\|\mathbb{E}[\mathbf{g}] - \nabla f(x)\|_1 \leq d^{-\Omega(1)}$. In particular, the randomness of the gradient computation is "entangled" with the randomness of the Markov chain. We use the classical analysis of ULD and ULD-RMM processes [36] to prove that the stochastic gradient will not significantly slow down the mixing of ULD processes, and that the error caused by the quantum gradient algorithm can be controlled.

**Quantum MALA** We next propose two quantum algorithms with lower query complexity than classical MALA, one with a Gaussian initial distribution and another with a warm-start distribution. The main technical tool we use is a quantum walk in continuous space.

The classical MALA (i.e., Metropolized HMC) starts from a Gaussian distribution $\mathcal{N}(0, L^{-1}I_d)$ and performs a leapfrog step in each iteration. It is well-known that the initial Gaussian state

$$|\rho_0\rangle = \int_{\mathbb{R}^d} \left(\frac{L}{2\pi}\right)^{d/4} e^{-\frac{L}{4}\|z-x^\star\|_2^2} |z\rangle \, \mathrm{d}z \tag{3.1}$$

can be efficiently prepared. We show that the quantum walk update operator

$$U := \int_{\mathbb{R}^d} \mathrm{d}x \int_{\mathbb{R}^d} \mathrm{d}y \, \sqrt{p_{x\to y}} |x\rangle\langle x| \otimes |y\rangle\langle 0| \tag{3.2}$$

can be efficiently implemented, where $p_{x\to y} := p(x, y)$ is the transition density from $x$ to $y$, and the density $p$ satisfies $\int_{\mathbb{R}^d} p(x, y) \, \mathrm{d}y = 1$ for any $x \in \mathbb{R}^d$.

**Lemma 3.1** (Informal version of Lemma C.6). *The continuous-space quantum walk operator corresponding to the MALA Markov chain can be implemented with $O(1)$ gradient and evaluation queries.*

In general, it is difficult to quantumly speed up the mixing time of a classical Markov chain, which is upper bounded by $O(\delta^{-1}\log(\rho_{\min}^{-1}))$, where $\delta$ is the spectral gap. However, [45] shows that a quadratic speedup is possible when following a sequence of *slowly-varying* Markov chains. More specifically, let $\rho_0, \ldots, \rho_r$ be the stationary distributions of the *reversible* Markov chains $\mathcal{M}_0, \ldots, \mathcal{M}_r$ and let $|\rho_0\rangle, \ldots, |\rho_r\rangle$ be the corresponding quantum states. Suppose $|\langle\rho_i|\rho_{i+1}\rangle| \geq p$ for all $i \in \{0, \ldots, r-1\}$, and suppose the spectral gaps of $\mathcal{M}_0, \ldots, \mathcal{M}_r$ are lower-bounded by $\delta$. Then we can prepare a quantum state $|\widetilde{\rho}_r\rangle$ that is $\epsilon$-close to $|\rho_r\rangle$ using $\tilde{O}(\delta^{-1/2}rp^{-1})$ quantum walk steps. To fulfill the slowly-varying condition, we consider an annealing process that goes from $\rho_0 = \mathcal{N}(0, L^{-1}I_d)$ to the target distribution $\rho_{M+1} = \rho$ in $M = \widetilde{O}(\sqrt{d})$ stages. At the $i$th stage, the stationary distribution is $\rho_i \propto e^{-f_i}$ with $f_i := f + \frac{1}{2}\sigma_i^{-2}\|x\|^2$. By properly choosing $\sigma_1 \leq \cdots \leq \sigma_M$, we prove that this sequence of Markov chains is slowly varying.

**Lemma 3.2** (Informal version of Lemma B.6). *If we take $\sigma_1^2 = \frac{\epsilon}{2dL}$ and $\sigma_{i+1}^2 = (1 + \frac{1}{\sqrt{d}})\sigma_i^2$, then for $0 \leq i \leq M$, we have $|\langle\rho_i|\rho_{i+1}\rangle| \geq \Omega(1)$.*

Combining Lemma 3.1, Lemma 3.2, and the effective spectral gap of MALA (Lemma C.7), we have:

**Theorem 3.2** (Informal version of Theorem C.7). *Let $\rho \propto e^{-f}$ be a d-dimensional log-concave distribution with $f$ satisfying* (1.1). *There is a quantum algorithm (Algorithm 1) that prepares a state $|\widetilde{\rho}\rangle$ with $\|\,|\widetilde{\rho}\rangle - |\rho\rangle\| \leq \epsilon$ using $\widetilde{O}(\kappa^{1/2}d)$ gradient and evaluation oracle queries.*

---

**Algorithm 1:** QUANTUMMALAFORLOG-CONCAVESAMPLING (Informal)

---

**Input:** Evaluation oracle $\mathcal{O}_f$, gradient oracle $\mathcal{O}_{\nabla f}$, smoothness parameter $L$, convexity parameter $\mu$

**Output:** Quantum state $|\widetilde{\rho}\rangle$ close to the stationary distribution state $\int_{\mathbb{R}^d} e^{-f(x)/2} \, \mathrm{d}|x\rangle$

1 Compute the cooling schedule parameters $\sigma_1, \ldots, \sigma_M$
2 Prepare the state $|\rho_0\rangle \propto \int_{\mathbb{R}^d} e^{-\frac{1}{4}\|x\|^2/\sigma_1^2} \, \mathrm{d}|x\rangle$
3 **for** $i \leftarrow 1, \ldots, M$ **do**
4     Construct $\mathcal{O}_{f_i}$ and $\mathcal{O}_{\nabla f_i}$ where $f_i(x) = f(x) + \frac{1}{2}\|x\|^2/\sigma_i^2$
5     Construct quantum walk update unitary $U$ with $\mathcal{O}_{f_i}$ and $\mathcal{O}_{\nabla f_i}$
6     Implement the quantum walk operator and the approximate reflection $\widetilde{R}_i$
7     Prepare $|\rho_i\rangle$ by performing $\frac{\pi}{3}$-amplitude amplification with $\widetilde{R}_i$ on the state $|\rho_{i-1}\rangle|0\rangle$
8 **return** $|\rho_M\rangle$

---

For the classical MALA with a Gaussian initial distribution, it was shown by [29] that the mixing time is at least $\widetilde{\Omega}(\kappa d)$. Theorem 3.2 quadratically reduces the $\kappa$ dependence.

Note that Algorithm 1 uses a first-order oracle, instead of the zeroth-order oracle used in the quantum ULD algorithms. The technical barrier to applying the quantum gradient algorithm (Lemma C.1) in the quantum MALA is to analyze the classical MALA with a stochastic gradient oracle. We currently do not know whether the "entangled randomness" dramatically increases the mixing time.

More technical details and proofs are provided in Appendix C.

## 4 Quantum Algorithm for Estimating Normalizing Constants

In this section, we apply our quantum log-concave sampling algorithms to the normalizing constant estimation problem. A very natural approach to this problem is via MCMC, which constructs a multi-stage annealing process and uses a sampler at each stage to solve a mean estimation problem. We show how to quantumly speed up these annealing processes and improve the query complexity of estimating normalizing constants.

**Quantum speedup for the standard annealing process** We first consider the standard annealing process for log-concave distributions, as already applied in the previous section. Recall that we pick parameters $\sigma_1 < \sigma_2 < \cdots < \sigma_M$ and construct a sequence of Markov chains with stationary distributions $\rho_i \propto e^{-f_i}$, where $f_i = f + \frac{1}{2\sigma_i^2}\|x\|^2$. Then, at the $i$th stage, we estimate the expectation

$$\mathbb{E}_{\rho_i}[g_i] \quad \text{where} \quad g_i = \exp\left(\frac{1}{2}(\sigma_i^{-2} - \sigma_{i+1}^{-2})\|x\|^2\right). \tag{4.1}$$

If we can estimate each expectation with relative error at most $O(\epsilon/M)$, then the product of these $M$ quantities estimates the normalizing constant $Z = \int_{\mathbb{R}^d} e^{-f(x)}\, \mathrm{d}x$ with relative error at most $\epsilon$.

For the mean estimation problem, [31] showed that when the relative variance $\frac{\mathbf{Var}_{\rho_i}[g_i]}{\mathbb{E}_{\rho_i}[g_i]^2}$ is constant, there is a quantum algorithm for estimating the expectation $\mathbb{E}_{\rho_i}[g_i]$ within relative error at most $\epsilon$ using $\widetilde{O}(1/\epsilon)$ quantum samples from the distribution $\rho_i$. Our annealing schedule satisfies the bounded relative variance condition. Therefore, by the quantum mean estimation algorithm, we improve the sampling complexity of the standard annealing process from $\widetilde{O}(M^2\epsilon^{-2})$ to $\widetilde{O}(M\epsilon^{-1})$.

To further improve the query complexity, we consider using the quantum MALAs developed in the previous section to generate samples. Observe that Algorithm 1 outputs a quantum state corresponding to some distribution that is close to $\rho_i$, instead of an individual sample. If we can estimate the expectation without destroying the quantum state, then we can reuse the state and evolve it for the $(i + 1)$st Markov chain. Fortunately, we can use non-destructive mean estimation to estimate the expectation and restore the initial states. A detailed error analysis of this algorithm can be found in [6, 22]. We first prepare $\widetilde{O}(M\epsilon^{-1})$ copies of initial states corresponding to the Gaussian distribution $\mathcal{N}(0, L^{-1}I_d)$. Then, for each stage, we apply the non-destructive mean estimation algorithm to estimate the expectation $\mathbb{E}_{\rho_i}[g_i]$ and then run quantum MALA to evolve the states $|\rho_i\rangle$ to $|\rho_{i+1}\rangle$. This gives our first quantum algorithm for estimating normalizing constants.

**Theorem 4.1** (Informal version of Theorem D.2). *Let $Z$ be the normalizing constant in (1.3). There is a quantum algorithm (Algorithm 2) that outputs an estimate $\widetilde{Z}$ with relative error at most $\epsilon$ using $\widetilde{O}\big(d^{3/2}\kappa^{1/2}\epsilon^{-1}\big)$ queries to the quantum gradient and evaluation oracles.*

**Quantum speedup for MLMC** Now we consider using multilevel Monte Carlo (MLMC) as the annealing process and show how to achieve quantum speedup. MLMC was originally developed by [23] for parametric integration; then [17] applied MLMC to simulate stochastic differential equations (SDEs). The idea of MLMC is natural: we choose a different number of samples at each stage based on the cost and variance of that stage.

To estimate normalizing constants, a variant of MLMC was proposed in [16]. Unlike the standard MLMC for bounding the mean-squared error, they upper bound the bias and the variance separately, and the analysis is technically difficult. The first quantum algorithm based on MLMC was subsequently developed by [2] based on the quantum mean estimation algorithm. Roughly speaking, the quantum algorithm can quadratically reduce the $\epsilon$-dependence of the sample complexity compared with classical MLMC.

---

**Algorithm 2:** QUANTUMMALAFORESTIMATINGNORMALIZINGCONSTANT (Informal)

**Input:** Evaluation oracle $\mathcal{O}_f$, gradient oracle $\mathcal{O}_{\nabla f}$

**Output:** Estimate $\widetilde{Z}$ of $Z$ with relative error at most $\epsilon$

1   $M \leftarrow \widetilde{O}(\sqrt{d})$, $K \leftarrow \widetilde{O}(\epsilon^{-1})$

2   Compute the cooling schedule parameters $\sigma_1, \ldots, \sigma_M$

3   **for** $j \leftarrow 1, \ldots, K$ **do**

4     Prepare the state $|\rho_{1,j}\rangle \propto \int_{\mathbb{R}^d} e^{-\frac{1}{4}\|x\|^2/\sigma_1^2}|x\rangle \mathrm{d}x$

5   $\widetilde{Z} \leftarrow (2\pi\sigma_1^2)^{d/2}$

6   **for** $i \leftarrow 1, \ldots, M$ **do**

7     $\widetilde{g}_i \leftarrow$ Non-destructive mean estimation for $g_i$ using $\{|\rho_{i,0}\rangle, \ldots, |\rho_{i,K}\rangle\}$

8     $\widetilde{Z} \leftarrow \widetilde{Z}\widetilde{g}_i$

9     **for** $j \leftarrow 1, \ldots, K$ **do**

10      $|\rho_{i+1,j}\rangle \leftarrow$ QUANTUMMALA$(\mathcal{O}_{f_{i+1}}, \mathcal{O}_{\nabla, f_{i+1}}, |\rho_{i,j}\rangle)$

11   **return** $\widetilde{Z}$

---

In this work, we apply the quantum accelerated MLMC (QA-MLMC) scheme [2] to simulate underdamped Langevin dynamics as the SDE. One challenge in using QA-MLMC is that $g_i$ in our setting is not Lipschitz. Fortunately, as suggested by [16], this issue can be resolved by truncating large $x$ and replacing $g_i$ by $h_i := \min\big\{g_i, \exp\big(\frac{(r_i^+)^2}{\sigma_i^2(1+\alpha^{-1})}\big)\big\}$, with the choice

$$\alpha = \widetilde{O}\left(\frac{1}{\sqrt{d}\log(1/\epsilon)}\right) \qquad r_i^+ = \mathbb{E}_{\rho_{i+1}}\|x\| + \Theta(\sigma_i\sqrt{(1+\alpha)\log(1/\epsilon)}) \qquad (4.2)$$

to ensure $\frac{h_i}{\mathbb{E}_{\rho_i}g_i}$ is $O(\sigma_i^{-1})$ Lipschitz. Furthermore, $\big|\mathbb{E}_{\rho_i}(h_i - g_i)\big| < \epsilon$ by Lemmas C.7 and C.8 in [16]. For simplicity, we regard $g_i$ as a Lipschitz continuous function in our main results. We present QA-MLMC in Algorithm 3, where the sampling algorithm A can be chosen to be quantum inexact ULD/ULD-RMM or quantum MALA.

---

**Algorithm 3:** QA-MLMC (Informal)

**Input:** Evaluation oracle $\mathcal{O}_f$, function $g$, error $\epsilon$, a quantum sampler $\mathrm{A}(x_0, f, \eta)$ for $\rho$

**Output:** An estimate of $\widetilde{R} = \mathbb{E}_\rho h$

1   $K \leftarrow \widetilde{O}(\epsilon^{-1})$

2   Compute the initial point $x_0$ and the step size $\eta_0$

3   Compute the number of samples $N_1, \ldots, N_K$

4   **for** $j \leftarrow 1, \ldots, K$ **do**

5     Let $\eta_j = \eta/2^{j-1}$

6     **for** $i \leftarrow 1, \ldots, N_j$ **do**

7      Sample $X_i^{\eta_j}$ by $\mathrm{A}(f, x_0, \eta_j)$, and sample $X_i^{\eta_j/2}$ by $\mathrm{A}(f, x_0, \eta_j/2)$

8      $\widetilde{G}_i^- \leftarrow$ QMEANEST$(\{g(X_i^{\eta_j})\}_{i\in[N_j]})$, and $\widetilde{G}_i^+ \leftarrow$ QMEANEST$(\{g(X_i^{\eta_j/2})\}_{i\in[N_j]})$

9   **return** $\widetilde{R} = \widetilde{G}_0 + \sum_{j=0}^K (\widetilde{G}_i^- - \widetilde{G}_i^+)$

---

This QA-MLMC framework reduces the $\epsilon$-dependence of the sampling complexity for estimating normalizing constants from $\epsilon^{-2}$ to $\epsilon^{-1}$ in both the ULD and ULD-RMM cases, as compared with the state-of-the-art classical results [16].

Using the quantum inexact ULD and ULD-RMM algorithms (Theorem 3.1) to generate samples, we obtain our second quantum algorithm for estimating normalizing constants (see Appendix D for proofs).

**Theorem 4.2** (Informal version of Theorem D.3 and Theorem D.4)**.** *Let $Z$ be the normalizing constant in (1.3). There exist quantum algorithms for estimating $Z$ with relative error at most $\epsilon$ using*

- *quantum inexact ULD with $\widetilde{O}(d^{3/2}\kappa^2\epsilon^{-1})$ queries to the evaluation oracle, and*

- *quantum inexact ULD-RMM with $\widetilde{O}((d^{7/6}\kappa^{7/6} + d^{4/3}\kappa)\epsilon^{-1})$ queries to the evaluation oracle.*

## 5  Quantum Lower Bound

Finally, we lower bound the quantum query complexity of normalizing constant estimation.

**Theorem 5.1.** *For any fixed positive integer $k$, given query access (1.4) to a function $f : \mathbb{R}^k \to \mathbb{R}$ that is 1.5-smooth and 0.5-strongly convex, the quantum query complexity of estimating the partition function $Z = \int_{\mathbb{R}^k} e^{-f(x)}\, \mathrm{d}x$ within multiplicative error $\epsilon$ with probability at least $2/3$ is $\Omega(\epsilon^{-\frac{1}{1+4/k}})$.*

The proof of our quantum lower bound is inspired by the construction in Section 5 of [16]. They consider a log-concave function whose value is negligible outside a hypercube centered at $0$. The interior of the hypercube is decomposed into cells of two types. The function takes different values on each type, and the normalizing constant estimation problem reduces to determining the number of cells of each type. Quantumly, we follow the same construction and reduce the cell counting problem to the *Hamming weight problem*: given an $n$-bit Boolean string and two integers $\ell < \ell'$, decide whether the Hamming weight (i.e., the number of ones) of this string is $\ell_1$ or $\ell_2$. This problem has a known quantum query lower bound [32], which implies the quantum hardness of estimating the normalizing constant. The full proof of Theorem 5.1 appears in Appendix E.

## Acknowledgements

AMC acknowledges support from the Army Research Office (grant W911NF-20-1-0015); the Department of Energy, Office of Science, Office of Advanced Scientific Computing Research, Accelerated Research in Quantum Computing program; and the National Science Foundation (grant CCF-1813814). TL was supported by a startup fund from Peking University, and the Advanced Institute of Information Technology, Peking University. JPL was supported by the National Science Foundation (grant CCF-1813814), an NSF Quantum Information Science and Engineering Network (QISE-NET) triplet award (DMR-1747426), a Simons Foundation award (No. 825053), and the Simons Quantum Postdoctoral Fellowship. RZ was supported by the University Graduate Continuing Fellowship from UT Austin.

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
