# A   Related Work

## A.1   Classical MCMC methods

Our quantum algorithms are inspired by a major class of classical MCMC algorithms based on *Langevin dynamics*. There has been extensive work on non-asymptotic error bounds for the mixing times of Langevin-type algorithms for sampling [8, 15, 28, 37, 47]. One commonly used type of algorithm is based on the mixing time of Langevin dynamics, including the underdamped Langevin diffusion process described by the stochastic differential equations

$$\mathrm{d}v_t = -\gamma v_t\, \mathrm{d}t - u\nabla f(x_t)\, \mathrm{d}t + \sqrt{2\gamma u}\, \mathrm{d}W_t \tag{A.1}$$

$$\mathrm{d}x_t = v_t\, \mathrm{d}t \tag{A.2}$$

with parameters $\gamma, u$, where $W_t \sim \mathcal{N}(0, t)$ is a standard Wiener process. The coefficients of (A.1) are Lipschitz continuous since $f$ is $L$-smooth; and the overdamped Langevin diffusion process

$$\mathrm{d}x_t = -u\nabla f(x_t)\, \mathrm{d}t + \sqrt{2u}\, \mathrm{d}W_t \tag{A.3}$$

is obtained by taking $\gamma \to \infty$ and $t \to t/\gamma$.

It can be shown that taking $\gamma = 2$ and $u = 1/L$, the stationary distribution of the underdamped Langevin diffusion (A.1) is proportional to $e^{-(f(x)+L\|v\|^2/2)}$, and the marginal distribution of $x$ is proportional to $e^{-f(x)}$. When $\gamma \to \infty$, the stationary distribution of the overdamped version (A.3) is proportional to $e^{-f(x)}$. The numerical discretization of (A.3) is used in unadjusted Langevin algorithms, while sampling algorithms based on the discretization of (A.1) can have a better dependence on $d$ and $\epsilon$.

We now introduce a few common classical sampling algorithms: the underdamped Langevin diffusion (ULD) method; the randomized midpoint method for underdamped Langevin diffusion (ULD-RMM), with the best known dependence on $d$; and the Metropolis adjusted Langevin algorithm (MALA), with the best known dependence on $\kappa$ and $\epsilon$. To simulate the random process in discrete time, ULD can be viewed as the first-order forward Euler discretization of the continuous process (A.1). In particular, ULD takes $\widetilde{O}(\kappa^2\sqrt{d}/\epsilon)$ steps to approximate the stationary distribution $e^{-f(x)}$ within $\epsilon$ in the Wasserstein 2-norm [10], where $\kappa$ is the condition number of $f$, and $d$ is the dimension. ULD-RMM approximates the integral of the random process (A.1) by randomly choosing the midpoint in the integral, which reduces the bias in the accumulation of the integration. As a more accurate approximation, ULD-RMM converges in the Wasserstein 2-distance $\epsilon$ with $\widetilde{O}\left(\frac{\kappa^{7/6}d^{1/6}}{\epsilon^{1/3}} + \frac{\kappa d^{1/3}}{\epsilon^{2/3}}\right)$ steps [37], a polynomial reduction in $\kappa, d, \epsilon$ over ULD. As an alternative approach, MALA also constructs the Euler discretization of (A.1), and then applies the Metropolis-Hastings acceptance/rejection step to ensure convergence to the correct stationary distribution. It was first shown that MALA converges in the total variation distance $\epsilon$ with $\widetilde{O}(\kappa d \max\{1, \kappa/d\}\log(\kappa d/\epsilon))$ steps for Gaussian initial distributions [8, 15]. Later, this result was improved to $\widetilde{O}(\kappa d \log(\kappa d/\epsilon))$ based on an improved non-asymptotic analysis of the mixing time [28]. For warm-start distributions, the complexity of MALA can be further reduced to $\widetilde{O}(\kappa d^{1/2}\log(\kappa d/\epsilon))$ [47]. Compared to ULD and ULD-RMM, this exponentially improves the dependence on $\epsilon$, and polynomially improves the dependence on $\kappa$, while it suffers from a worse dependence on $d$. We introduce the algorithms and complexities of ULD and ULD-RMM in Appendix B.1, and introduce these results of MALA in Appendix C.2.1.

For the task of estimating the normalizing constant (1.3), the state-of-the-art classical results are given by [16]. That work applies the classical sampling algorithms described above with an annealing strategy. The normalizing constant is estimated by a sequence of telescoping sums, each of which can be approximated by a Monte Carlo method that samples from a log-concave distribution. We introduce this annealing procedure in Appendix B.2. Reference [16] employed the mixing time of MALA for Gaussian initial distributions developed by [8, 15] with the annealing procedure, achieving the overall complexity $\widetilde{O}\left(\frac{\kappa d^2}{\epsilon^2}\max\{1, \kappa/d\}\right)$ for estimating the normalizing constant. They also combined ULD and ULD-RMM with the annealing and the multilevel Monte Carlo (MLMC) method to achieve complexities of $\widetilde{O}\left(\frac{\kappa^2 d^{3/2}}{\epsilon^2}\right)$ and $\widetilde{O}\left(\frac{\kappa^{7/6}d^{7/6}}{\epsilon^2} + \frac{\kappa d^{4/3}}{\epsilon^2}\right)$, respectively. Here MLMC, introduced in Appendix D.2, is utilized to resolve the worse dependence on $\epsilon$ in

ULD and ULD-RMM, resulting in the same $\widetilde{O}(1/\epsilon^2)$ scaling of the error compared to the annealing with MALA. Annealing with MLMC and ULD/ULD-RMM also has a better dependence on $d$ over annealing with MALA, while they suffer from a worse dependence on $\kappa$.

## A.2 Quantum computing

Previous literature developed alternative approaches to generating quantum states corresponding to classical probability distributions on a quantum computer, sometimes referred to as quantum sampling (or qsampling) from a distribution. References [49], [19], and [26] propose direct state generation approaches using controlled rotations. However, this approach is limited to the regime in which the distribution is efficiently integrable. As an alternative, [1] develops an adiabatic approach to qsampling. They apply adiabatic evolution techniques to qsample the stationary distributions of a sequence of slowly varying Markov chains, a technique referred to as quantum simulated annealing (QSA) in subsequent literature [22, 38, 39, 45, 48]. The time complexity of Aharanov and Ta-Shma's approach is $O(1/\delta)$ as a function of the spectral gap $\delta$, comparable to the running time of analogous classical sampling methods. Reference [45] adopted Szegedy's quantum walks [41] and amplitude amplification [5] to improve the time complexity of this qsampling procedure to $O(1/\sqrt{\delta})$, achieving a quadratic speedup in the spectral gap. As a generalization, [42] proposes a quantum Metropolis sampling method that extends qsampling to quantum Hamiltonians, with time complexity $O(1/\delta)$. Reference [48] combines quantum Metropolis sampling with QSA to achieve time complexity $O(1/\sqrt{\delta})$. Another alternative approach is quantum rejection sampling [30, 33, 44], which provides a method for transforming an initial superposition of desired and undesired states into the desired state using amplitude amplification. Reference [44] employs semi-classical Bayesian updating to achieve time complexity $O(1/\sqrt{\epsilon})$ as a function of the approximation error $\epsilon$. The quantum rejection sampling approach is generally less efficient than the QSA approach, as the latter can achieve $O(\log 1/\epsilon)$ by choosing proper slowly varying MCs that mix rapidly.

Previous quantum computing literature on partition function estimation mainly focused on discrete systems with

$$Z(\beta) = \sum_{x \in \Omega} e^{-\beta H(x)}, \tag{A.4}$$

where $\beta$ is an inverse temperature and $H$ is a Hamiltonian function of $x$ over a finite state space $\Omega$. The space $\Omega$ is usually assumed to be a simple discrete set, such as $\{0, 1\}^n$, and $H$ is assumed to be a sum of local terms. For instance, [31] considers $H$ taking integer values $\{0, 1, \ldots, n\}$, and [22] assumes $0 \le H(x) \le n$ for all $x$.

To estimate $Z = Z(\infty)$ in (A.4), [31] considers a classical Chebyshev cooling schedule $0 = \beta_0 < \beta_1 < \ldots \beta_l = \infty$ for $Z$ with the length $l = O(\sqrt{\log|\Omega|}\log\log|\Omega|)$ [40]. Reference [31] applies fast qsampling algorithms to estimate $Z$ with $\widetilde{O}(l^2/\sqrt{\delta}\epsilon) = \widetilde{O}(\log|\Omega|/\sqrt{\delta}\epsilon)$ quantum walk steps to sample from Gibbs distributions $\pi_i(x) = \frac{1}{Z(\beta_i)}e^{-\beta_i H(x)}$, whereas a corresponding classical algorithm takes $\widetilde{O}(l^2/\delta\epsilon^2) = \widetilde{O}(\log|\Omega|/\delta\epsilon^2)$ random walk steps. Here $\epsilon$ denotes the relative error for estimating $Z$, and $\delta$ denotes the spectral gap of the Markov chains with stationary distributions $\pi_i(x)$. Reference [31] also points out that this quantum algorithm relies on classical Markov chain Monte Carlo for computing the Chebyshev cooling schedule, introducing an overhead of $\widetilde{O}(\log|\Omega|/\delta)$ [40]. Hence, the overall cost is $\widetilde{O}(\log|\Omega|/\sqrt{\delta}(\epsilon + \sqrt{\delta}))$, a quadratic reduction with respect to $\epsilon$ over classical algorithms. Reference [22] develops a fully quantized version of the Chebyshev cooling schedule that only requires additional cost $\widetilde{O}(\log|\Omega|/\sqrt{\delta})$. This results in overall cost $\widetilde{O}(\log|\Omega|/\sqrt{\delta}\epsilon)$, a quadratic speedup in terms of $\delta$ over [31] and classical algorithms. Reference [4] constructs a shorter Chebyshev cooling schedule by using a paired-product estimator with length $l = O(\sqrt{\log|\Omega|})$, eliminating the $l = O(\log\log|\Omega|)$ factors in the previous schedule [40]. Reference [4] develops a fully quantized version of this shorter schedule, almost matching the same overall cost $\widetilde{O}(\log|\Omega|/\sqrt{\delta}(\epsilon + \sqrt{\delta}))$ of [22].

Estimating the partition function of a discrete system corresponds to a discrete counting problem, with applications such as counting colorings or matchings of a graph and estimating statistics of Ising models, while estimating partition functions of continuous systems is relevant to the volume estimation problem.

# B Tools from Classical MCMC Algorithms

## B.1 ULD and ULD-RMM

We now describe underdamped Langevin diffusion (ULD) and the randomized midpoint method for underdamped Langevin diffusion (ULD-RMM), as introduced in [16] with Lipschitz continuous constants. We consider the underdamped Langevin diffusion with parameters $\gamma, u$:

$$dv_t = -\gamma v_t \, dt - \nabla f(x_t) \, dt + \sqrt{2\gamma u} \, dW_t, \tag{B.1}$$

$$dx_t = v_t \, dt, \tag{B.2}$$

The discrete dynamics of ULD can be described by

$$dv_t^h = -\gamma v_t^h \, dt - u \nabla f(x_{\lfloor t/h \rfloor h}^h) \, dt + \sqrt{2\gamma u} \, dW_t, \tag{B.3}$$

$$dx_t^h = v_t^h \, dt. \tag{B.4}$$

According to [16], taking $\gamma = 2$ and $u = 1/L$, the explicit discrete-time update of ULD is integrated as

$$v_{t+h}^h = e^{-2h} v_t^h + \frac{1}{2L}(1 - e^{-2h})\nabla f(x_t^h) + \frac{2}{\sqrt{L}}W_{1,t}^h, \tag{B.5}$$

$$x_{t+h}^h = x_t^h + \frac{1}{2}(1 - e^{-2h})v_t^h + \frac{1}{2L}[h - (1 - e^{-2h})]\nabla f(x_t^h) + \frac{1}{\sqrt{L}}W_{2,t}^h, \tag{B.6}$$

where

$$W_{1,t}^h = \int_0^h e^{2(s-h)}dB_{t+s}, \tag{B.7}$$

$$W_{2,t}^h = \int_0^h (1 - e^{2(s-h)})dB_{t+s}. \tag{B.8}$$

$W_{1,t}^h$ and $W_{2,t}^h$ can be obtained by sampling the $d$-dimensional standard Brownian motion $B_t$.

The ULD algorithm is presented in Algorithm 4. The convergence of ULD has been established by Theorem 1 of [10], which was restated by Theorem C.3 of [16] as follows.

---

**Algorithm 4:** Underdamped Langevin Dynamics (ULD)

---
**Input:** Function $f$, step size $h$, time $T$, and a sample $x_0$ from a starting distribution $\rho_0$
**Output:** Sequence $x_h^h, x_{2h}^h, \ldots, x_{\lfloor T \rfloor + 1}^h$

1 Compute $x_0^h \leftarrow x_0$
2 **for** $t = 0, h, \ldots, \lfloor T \rfloor$ **do**
3     Draw $W_{1,t}^h = \int_0^h e^{2(s-h)}dB_{t+s}, W_{2,t}^h = \int_0^h (1 - e^{2(s-h)})dB_{t+s}$
4     Compute $v_{t+h}^h = e^{-2h}v_t^h + \frac{1}{2L}(1 - e^{-2h})\nabla f(x_t^h) + \frac{2}{\sqrt{L}}W_{1,t}^h,$
      $x_{t+h}^h = x_t^h + \frac{1}{2}(1 - e^{-2h})v_t^h + \frac{1}{2L}[h - (1 - e^{-2h})]\nabla f(x_t^h) + \frac{1}{\sqrt{L}}W_{2,t}^h$

---

**Lemma B.1** (Theorem 1 of [10]). *Assume the target distribution $\rho$ is strongly log-concave with $L$-smooth and $\mu$-strongly convex negative log-density. Let $\rho_n$ be the distribution of the underdamped Langevin diffusion with the initial point $x_0$ satisfying $\|x_0 - x^*\| \leq D$, step size $h \leq \frac{\epsilon}{104\kappa}\sqrt{\frac{1}{d/\mu + D^2}}$, and time $T \geq \frac{\kappa}{2}\log\left(\frac{24\sqrt{d/\mu + D^2}}{\epsilon}\right)$. Then ULD achieves*

$$\mathbb{E}\left(\|\widehat{X}_n - X_T\|^2\right) \leq \widetilde{O}\left(\frac{d^2\kappa^2 h^2}{\mu}\right), \tag{B.9}$$

$$W_2(\rho_n, \rho) \leq \epsilon, \tag{B.10}$$

*using*

$$\frac{T}{h} = \widetilde{\Theta}\left(\frac{\kappa^2\sqrt{d}}{\epsilon}\right) \tag{B.11}$$

*queries to $\nabla f$.*

According to [16], the explicit discrete-time update of ULD-RMM is integrated as

$$v_{t+h}^h = e^{-2h}v_t^h + \frac{h}{L}e^{-2(1-\alpha)h}\nabla f(y_t^h) + \frac{2}{\sqrt{L}}W_{1,t}^h, \tag{B.12}$$

$$x_{t+h}^h = x_t^h + \frac{1}{2}(1 - e^{-2h})v_t^h + \frac{h}{2L}(1 - e^{-2(1-\alpha)h})\nabla f(y_t^h) + \frac{1}{\sqrt{L}}W_{2,t}^h, \tag{B.13}$$

$$y_{t+h}^h = x_t^h + \frac{1}{2}(1 - e^{-2\alpha h})v_t^h + \frac{1}{2L}[\alpha h - (1 - e^{-2\alpha h})]\nabla f(x_t^h) + \frac{1}{\sqrt{L}}W_{3,t}^h, \tag{B.14}$$

where

$$W_{1,t}^h = \int_0^h e^{2(s-h)}dB_{t+s}, \tag{B.15}$$

$$W_{2,t}^h = \int_0^h (1 - e^{2(s-h)})dB_{t+s}, \tag{B.16}$$

$$W_{3,t}^h = \int_0^{\alpha h} (1 - e^{2(s-h)})dB_{t+s}. \tag{B.17}$$

$W_{1,t}^h$, $W_{2,t}^h$, and $W_{3,t}^h$ can be obtained by sampling the $d$-dimensional standard Brownian motion $B_t$.

The ULD-RMM algorithm is presented in Algorithm 5. The convergence of ULD-RMM has been established by Theorem 3 of [37], which was restated by Theorem C.5 of [16] as follows.

---

**Algorithm 5:** Underdamped Langevin Dynamics with Randomized Midpoint Method (ULD-RMM)

---

**Input:** Function $f$, step size $h$, time $T$, and a sample $x_0$ from a starting distribution $\rho_0$
**Output:** Sequence $x_h^h, x_{2h}^h, \ldots, x_{\lfloor T \rfloor +1}^h$

1 Compute $x_0^h \leftarrow x_0$, $y_0^h \leftarrow x_0$
2 **for** $t = 0, h, \ldots, \lfloor T \rfloor$ **do**
3      Draw $W_{1,t}^h = \int_0^h e^{2(s-h)}dB_{t+s}$, $W_{2,t}^h = \int_0^h (1 - e^{2(s-h)})dB_{t+s}$,
       $W_{3,t}^h = \int_0^{\alpha h}(1 - e^{2(s-h)})dB_{t+s}$
4      Compute $v_{t+h}^h = e^{-2h}v_t^h + \frac{h}{L}e^{-2(1-\alpha)h}\nabla f(y_t^h) + \frac{2}{\sqrt{L}}W_{1,t}^h$,
       $x_{t+h}^h = x_t^h + \frac{1}{2}(1 - e^{-2h})v_t^h + \frac{h}{2L}(1 - e^{-2(1-\alpha)h})\nabla f(y_t^h) + \frac{1}{\sqrt{L}}W_{2,t}^h$,
       $y_{t+h}^h = x_t^h + \frac{1}{2}(1 - e^{-2\alpha h})v_t^h + \frac{1}{2L}[\alpha h - (1 - e^{-2\alpha h})]\nabla f(x_t^h) + \frac{1}{\sqrt{L}}W_{3,t}^h$

---

**Lemma B.2** (Theorem 3 of [37]). *Assume the target distribution $\rho$ is strongly log-concave with $L$-smooth and $\mu$-strongly convex negative log-density. Let $\rho_n$ be the distribution of the randomized midpoint method for underdamped Langevin diffusion with the initial point $x_0$, step size $h \leq \min\left\{\frac{\epsilon^{1/3}\mu^{1/6}}{\kappa^{1/6}d^{1/6}\log^{1/6}\left(\frac{\sqrt{d/\mu}}{\epsilon}\right)}, \frac{\epsilon^{2/3}\mu^{1/3}}{d^{1/3}\log^{1/3}\left(\frac{\sqrt{d/\mu}}{\epsilon}\right)}\right\}$, and time $T \geq 2\kappa \log\left(\frac{20d/\mu}{\epsilon^2}\right)$. Then ULD-RMM achieves*

$$\mathbb{E}\left(\|\widehat{X}_n - X_T\|^2\right) \leq \widetilde{O}\left(\frac{d\kappa h^6}{\mu} + \frac{dh^3}{\mu}\right), \tag{B.18}$$

$$W_2(\rho_n, \rho) \leq \epsilon, \tag{B.19}$$

*using*

$$\frac{2T}{h} = \widetilde{\Theta}\left(\frac{\kappa^{7/6}d^{1/6}}{\epsilon^{1/3}} + \frac{\kappa d^{1/3}}{\epsilon^{2/3}}\right) \tag{B.20}$$

*queries to $\nabla f$.*

## B.2 Annealing for Estimating the Normalizing Constant

Having described the sampling procedure for a log-concave function, we now move to the problem of estimating the normalizing constant

$$Z = \int_{x\in\mathbb{R}^d} e^{-f(x)}dx. \tag{B.21}$$

We consider a sequence of auxiliary distributions, given by

$$f_i(x) = \frac{1}{2} \frac{\|x\|^2}{\sigma_i^2} + f(x) \tag{B.22}$$

for $i \in [M]$, where $\sigma_1 \leq \sigma_2 \leq \cdots \leq \sigma_M$. We define $\sigma_{M+1} = \infty$ and $f_{M+1} = f$ for convenience. We consider the sequence of distributions

$$\rho_i(\mathrm{d}x) = Z_i^{-1} e^{-f_i(x)} \mathrm{d}x, \tag{B.23}$$

where $Z_i$ is the normalizing constant

$$Z_i = \int_{x \in \mathbb{R}^d} e^{-f_i(x)} \mathrm{d}x. \tag{B.24}$$

Then $Z$ is estimated by the telescoping product

$$Z = Z_{M+1} = Z_1 \prod_{i=1}^{M} \frac{Z_{i+1}}{Z_i}. \tag{B.25}$$

In (B.25), we first approximate $Z_1$ by the normalizing constant of the Gaussian distribution with variance $\sigma_1^2$, which is bounded by the following lemma.

**Lemma B.3** (Lemma 3.1 of [16]). *Letting $\sigma_1^2 = \frac{\epsilon}{2dL}$, we have*

$$\left(1 - \frac{\epsilon}{2}\right) \int_{x \in \mathbb{R}^d} e^{-\frac{1}{2} \frac{\|x\|^2}{\sigma_1^2}} \mathrm{d}x \leq Z_1 \leq \int_{x \in \mathbb{R}^d} e^{-\frac{1}{2} \frac{\|x\|^2}{\sigma_1^2}} \mathrm{d}x. \tag{B.26}$$

We then approximate $\frac{Z_{i+1}}{Z_i}$ by sampling the distribution $\rho_i$, with

$$\frac{Z_{i+1}}{Z_i} = \mathbb{E}_{\rho_i}(g_i), \tag{B.27}$$

where

$$g_i = \exp\left(\frac{1}{2}\left(\frac{1}{\sigma_i^2} - \frac{1}{\sigma_{i+1}^2}\right)\|x\|^2\right). \tag{B.28}$$

If $X_i^{(1)}, X_i^{(2)}, \dots, X_i^{(K)}$ are i.i.d. samples generated according to the distribution $\rho_i$, then

$$\frac{Z_{i+1}}{Z_i} \approx \frac{1}{K} \sum_{k=1}^{K} g_i(X_i^{(k)}). \tag{B.29}$$

For the sequence of $\sigma_i^2$ with the annealing schedule $\frac{\sigma_{i+1}^2}{\sigma_i^2} = 1 + \alpha$, we aim to bound the relative variance of $\frac{Z_{i+1}}{Z_i}$. First, for $\frac{Z_{M+1}}{Z_M}$, we have the following lemma.

**Lemma B.4** (Lemma 3.2 of [16]). *For $\sigma_M^2 \geq \frac{2}{\mu}$, we have*

$$\frac{\mathbb{E}_{\rho_M}(g_M^2)}{\mathbb{E}_{\rho_M}(g_M)^2} \leq \exp\left(\frac{4d}{\mu \sigma_M^4}\right). \tag{B.30}$$

When $\sigma_M^2 \geq \frac{2\sqrt{d}}{\mu}$, and assuming $\mu < 1$, we have $\frac{\mathbb{E}_{\rho_M}(g_M^2)}{\mathbb{E}_{\rho_M}(g_M)^2} \leq e$.

Second, for $\frac{Z_{i+1}}{Z_i}$ with $i \in [M-1]$, the relative variance of can be bounded by the following lemma.

**Lemma B.5** ([Lemma 3.3 of [16]). *Let $\rho$ be a log-concave distribution. For $\alpha \leq \frac{1}{2}$, we have*

$$\frac{\mathbb{E}_{\rho_i}(g_i^2)}{\mathbb{E}_{\rho_i}(g_i)^2} = \frac{\mathbb{E}_\rho\left[\exp\left(-\frac{1+\alpha}{2} \frac{\|x\|^2}{\sigma^2}\right)\right] \cdot \mathbb{E}_\rho\left[\exp\left(-\frac{1-\alpha}{2} \frac{\|x\|^2}{\sigma^2}\right)\right]}{\mathbb{E}_\rho\left[\exp\left(-\frac{1}{2} \frac{\|x\|^2}{\sigma^2}\right)\right]^2} \leq \exp\left(4\alpha^2 d\right). \tag{B.31}$$

Therefore, if we choose the annealing schedule

$$\frac{\sigma_{i+1}^2}{\sigma_i^2} = 1 + \frac{1}{\sqrt{d}}, \tag{B.32}$$

then $\frac{\mathbb{E}_{\rho_i}(g_i^2)}{\mathbb{E}_{\rho_i}(g_i)^2} \leq e^4$.

The estimate of the normalizing constant (1.3) relies on the above annealing framework and the sampling algorithms for the log-concave distribution $\rho_i$ including ULD, ULD-RMM, and MALA. In the following sections, we discuss the quantum speedup for (1.3) using MALA and annealing, and using multilevel ULD/ULD-RMM and annealing.

## B.3  Annealing Markov chains are slowly varying

The goal of this subsection is to prove the following lemma.

**Lemma B.6** (Slowly varying MCs). *Let $f_0(x) = \frac{\|x\|^2}{2\sigma_1^2}$ and let $\mathrm{d}\pi_0 = (2\pi\sigma_1^2)^{d/2} \cdot e^{-f_0(x)}\mathrm{d}x$ be the Gaussian distribution. For $i \in \{1, \ldots, M\}$, let $f_i(x) = f(x) + \frac{\|x\|^2}{2\sigma_i^2}$ and let $\mathrm{d}\pi_i = Z_i^{-1}e^{-f_i(x)}\mathrm{d}x$ be its stationary distribution. Let $f_{M+1}(x) = f(x)$ and let $\mathrm{d}\pi_{M+1}$ be the target log-concave distribution. Define the qsample state*

$$|\pi_i\rangle = \int_\Omega \mathrm{d}x \sqrt{\pi_i(x)}|x\rangle \quad \forall 0 \leq i \leq M+1. \tag{B.33}$$

*Then, for $0 \leq i \leq M$, we have*

$$|\langle\pi_i|\pi_{i+1}\rangle| \geq \Omega(1). \tag{B.34}$$

*Proof.* First, we consider the case when $i = 0$. Note that $|\langle\pi_0|\pi_1\rangle|$ can be written as

$$|\langle\pi_0|\pi_1\rangle| = \int_\Omega \mathrm{d}x \cdot (2\pi\sigma_1^2)^{-d/4}e^{-\frac{1}{2}f_0(x)} \cdot Z_1^{-1/2}e^{-\frac{1}{2}f_1(x)} \tag{B.35}$$

$$= \frac{\int_\Omega e^{-\frac{1}{2}f(x)-\frac{\|x\|^2}{2\sigma_1^2}}\mathrm{d}x}{(2\pi\sigma_1^2)^{d/4} \cdot \sqrt{Z_1}}. \tag{B.36}$$

Since $0 \leq f(x) \leq \frac{1}{2}L\|x\|^2$, the numerator can be lower bounded by

$$\int_\Omega e^{-\frac{1}{2}f(x)-\frac{\|x\|^2}{2\sigma_1^2}}\mathrm{d}x \geq \int_\Omega e^{-\frac{1}{2}(L/2+\sigma_1^{-2})\|x\|^2}\mathrm{d}x = \left(2\pi(L/2+\sigma_1^{-2})^{-1}\right)^{d/2} \tag{B.37}$$

and the denominator can be upper bounded by

$$(2\pi\sigma_1^2)^{d/4} \cdot \sqrt{\int_\Omega e^{-f(x)-\frac{1}{2}\|x\|^2/\sigma_1^2}\mathrm{d}x} \leq (2\pi\sigma_1^2)^{d/4} \cdot \sqrt{\int_\Omega e^{-\frac{1}{2}\|x\|^2/\sigma_1^2}\mathrm{d}x} = (2\pi\sigma_1^2)^{d/2}. \tag{B.38}$$

Therefore

$$|\langle\pi_0|\pi_1\rangle| \geq \frac{\left(2\pi(L/2+\sigma_1^{-2})^{-1}\right)^{d/2}}{(2\pi\sigma_1^2)^{d/2}} = (1+\sigma_1^2L/2)^{-d/2} \geq e^{-\sigma_1^2dL/4}. \tag{B.39}$$

By our choice of $\sigma_1^2 = \frac{\epsilon}{2dL}$, we have $|\langle\pi_0|\pi_1\rangle| \geq e^{-\epsilon/8} = \Omega(1)$.

Now consider the case where $1 \leq i \leq M-1$. The inner product between $|\pi_i\rangle$ and $|\pi_{i+1}\rangle$ can be written as

$$|\langle\pi_i|\pi_{i+1}\rangle| = \int_\Omega \mathrm{d}x \cdot Z_i^{-1/2}e^{-\frac{1}{2}f_i(x)} \cdot Z_{i+1}^{-1/2}e^{-\frac{1}{2}f_{i+1}(x)} \tag{B.40}$$

$$= \frac{\int_\Omega e^{-f(x)-\frac{\|x\|^2}{4}(\sigma_i^{-2}+\sigma_{i+1}^{-2})}\mathrm{d}x}{\sqrt{Z_iZ_{i+1}}}. \tag{B.41}$$

Let $\sigma^2 = \sigma_{i+1}^2$ and $\sigma^2/(1+\alpha) = \sigma_i^2$. Also, let $\rho$ be the log-concave distribution $\rho(\mathrm{d}x) = Z^{-1}e^{-f(x)}\mathrm{d}x$. Then we have

$$\int_\Omega e^{-f(x) - \frac{\|x\|^2}{4}(\sigma_i^{-2} + \sigma_{i+1}^{-2})} \mathrm{d}x = Z \cdot \mathbb{E}_\rho\left[e^{-\frac{1+\alpha/2}{2\sigma^2}\|x\|^2}\right]. \tag{B.42}$$

Similarly,

$$Z_i = Z \cdot \mathbb{E}_\rho\left[e^{-\frac{1+\alpha}{2\sigma^2}\|x\|^2}\right] \quad \text{and} \quad Z_{i+1} = Z \cdot \mathbb{E}_\rho\left[e^{-\frac{1}{2\sigma^2}\|x\|^2}\right]. \tag{B.43}$$

Hence,

$$|\langle \pi_i | \pi_{i+1} \rangle| = \frac{\mathbb{E}_\rho\left[e^{-\frac{1+\alpha/2}{2\sigma^2}\|x\|^2}\right]}{\mathbb{E}_\rho\left[e^{-\frac{1+\alpha}{2\sigma^2}\|x\|^2}\right]^{1/2} \cdot \mathbb{E}_\rho\left[e^{-\frac{1}{2\sigma^2}\|x\|^2}\right]^{1/2}}. \tag{B.44}$$

Let $\alpha' := \frac{\alpha}{\alpha+2}$ and $\sigma'^2 := \frac{\sigma^2}{1+\alpha/2}$. Then

$$|\langle \pi_i | \pi_{i+1} \rangle| = \frac{\mathbb{E}_\rho\left[e^{-\frac{1}{2\sigma'^2}\|x\|^2}\right]}{\mathbb{E}_\rho\left[e^{-\frac{1+\alpha'}{2\sigma'^2}\|x\|^2}\right]^{1/2} \cdot \mathbb{E}_\rho\left[e^{-\frac{1-\alpha'}{2\sigma'^2}\|x\|^2}\right]^{1/2}} \tag{B.45}$$

$$\geq e^{-2\alpha'^2 d}, \tag{B.46}$$

where the last step follows from Lemma B.5. Since we choose $\alpha = d^{-1/2}$, we have $\alpha' = \frac{1}{1+2\sqrt{d}} = O(d^{-1/2})$, which implies that $e^{-2\alpha^2 d} = \Omega(1)$.

Finally, we consider the case where $i = M$. The inner product can be written as

$$|\langle \pi_M | \pi_{M+1} \rangle| = \frac{\int_\Omega \mathrm{d}x \cdot e^{-\frac{1}{2}f(x) - \frac{1}{4}\|x\|^2/\sigma_M^2} \cdot e^{-\frac{1}{2}f(x)}}{\sqrt{Z_M}\sqrt{Z}} \tag{B.47}$$

$$= \frac{\int_\Omega \mathrm{d}x \cdot e^{-f(x) - \frac{1}{4}\|x\|^2/\sigma_M^2}}{\sqrt{Z_M}\sqrt{Z}}. \tag{B.48}$$

Let $\rho'$ be a log-concave distribution with density proportional to $e^{-f(x) - \frac{1}{4}\|x\|^2/\sigma_M^2}$. Then

$$\frac{\int_\Omega \mathrm{d}x \cdot e^{-f(x) - \frac{1}{4}\|x\|^2/\sigma_M^2}}{\sqrt{Z_M}\sqrt{Z}} = \mathbb{E}_{\rho'}\left[e^{-\frac{1}{4}\|x\|^2/\sigma_M^2}\right]^{-1/2} \cdot \mathbb{E}_{\rho'}\left[e^{\frac{1}{4}\|x\|^2/\sigma_M^2}\right]^{-1/2} \tag{B.49}$$

$$\geq e^{-\frac{d}{2\mu\sigma_M^4}}, \tag{B.50}$$

where the second step follows from the proof of Lemma B.4 in [16]. Since we take $\sigma_M^2 = \Theta(\frac{\sqrt{d}}{\mu})$, we find that $|\langle \pi_M | \pi_{M+1} \rangle| \geq e^{-\Theta(1)} = \Omega(1)$.

Combining the three cases, the proof is complete. $\qquad\square$

## C  Quantum Algorithm for Log-Concave Sampling: Details

In this section, we provide several quantum algorithms for sampling log-concave distributions. In Appendix C.1, we show that the classical underdamped Langevin diffusion (ULD) and the randomized midpoint method for underdamped Langevin diffusion (ULD-RMM) can be improved by replacing the first-order oracle by the zeroth-order quantum oracle, while achieving the same efficiency and accuracy guarantees. In Appendix C.2, we show that the Metropolis adjusted Langevin algorithm (MALA) can be quantumly sped up in terms of query complexity, for both Gaussian initial distributions and warm-start distributions.

## C.1 Quantum Inexact ULD and ULD-RMM

In the quantum setting, we can estimate $\nabla f(x)$ by using Jordan's algorithm with queries to the quantum zeroth-order evaluation oracle (1.4). The following lemma provides an $\ell_1$-error guarantee.

**Lemma C.1** (Lemma 2.3 in [7]). *Let $f$ be a convex, $L_0$-Lipschitz continuous function that is specified by an evaluation oracle with error at most $\epsilon$. Suppose $f$ is $L$-smooth in $B_\infty(x, 2\sqrt{\epsilon/L})$. Let*

$$\widetilde{g} = \mathsf{SmoothQuantumGradient}(f, \epsilon, L_0, L, x). \tag{C.1}$$

*Then for any $i \in [d]$, we have $|\widetilde{g}_i| \leq L_0$ and $\mathbb{E}|\widetilde{g}_i - \nabla f(x)_i| \leq 3000\sqrt{d\epsilon L}$; hence*

$$\mathbb{E}\|\widetilde{g} - \nabla f(x)\|_1 \leq 3000 d^{1.5}\sqrt{\epsilon L}. \tag{C.2}$$

*If $L_0$, $1/L$, and $1/\epsilon$ are $\mathrm{poly}(d)$, the $\mathsf{SmoothQuantumGradient}$ algorithm uses $O(1)$ queries to the quantum evaluation oracle and $\widetilde{O}(d)$ gates.*

We then introduce inexact ULD and ULD-RMM by using a stochastic zeroth-order oracle as follows.

**Lemma C.2** (Theorem 2.2 of [36]). *Let $\rho_n$ be the distribution of the underdamped Langevin diffusion with the initial point $x_0$ satisfying $\|x_0 - x^*\| \leq D$, step size $h \leq \frac{\epsilon}{104\kappa}\sqrt{\frac{1}{d/\mu + D^2}}$, and time $T \geq \frac{\kappa}{2}\log\left(\frac{24\sqrt{d/\mu + D^2}}{\epsilon}\right)$. Assume there is a stochastic zeroth-order oracle that provides an unbiased evaluation of $\nabla f(x)$ with bounded variance $\mathbb{E}\|\widetilde{g} - \nabla f(x)\|^2 \leq \sigma^2$. Then inexact ULD achieves $W_2(\rho_n, \rho) \leq \epsilon$ using*

$$\frac{T}{h} = \widetilde{\Theta}\left(\frac{\kappa^2\sqrt{d}}{\epsilon}\right) \tag{C.3}$$

*iterations and*

$$b = \frac{d^{1.5}\max\{1, \sigma^2\}}{\epsilon} \tag{C.4}$$

*queries to the zeroth-order oracle per iteration. The total number of calls is $\frac{bT}{h}$.*

**Lemma C.3** (Theorem 2.3 of [36]). *Let $\rho_n$ be the distribution of the randomized midpoint method for underdamped Langevin diffusion with the initial point $x_0$, step size $h \leq \min\left\{\frac{\epsilon^{1/3}\mu^{1/6}}{\kappa^{1/6}d^{1/6}\log^{1/6}\left(\frac{\sqrt{d/\mu}}{\epsilon}\right)}, \frac{\epsilon^{2/3}\mu^{1/3}}{d^{1/3}\log^{1/3}\left(\frac{\sqrt{d/\mu}}{\epsilon}\right)}\right\}$, and time $T \geq 2\kappa\log\left(\frac{20d/\mu}{\epsilon^2}\right)$. Assume there is a stochastic zeroth-order oracle that provides an unbiased evaluation of $\nabla f(x)$ with bounded variance $\mathbb{E}\|\widetilde{g} - \nabla f(x)\|^2 \leq \sigma^2$. Then inexact ULD-RMM achieves $W_2(\rho_n, \rho) \leq \epsilon$ using*

$$\frac{2T}{h} = \widetilde{\Theta}\left(\frac{\kappa^{7/6}d^{1/6}}{\epsilon^{1/3}} + \frac{\kappa d^{1/3}}{\epsilon^{2/3}}\right) \tag{C.5}$$

*iterations and*

$$b = \frac{d\kappa}{h^3} \tag{C.6}$$

*queries to the zeroth-order oracle per iteration. The total number of calls is $\frac{bT}{h}$.*

As a quantum counterpart, we are able to reduce the number of queries from $O(b)$ to $O(1)$ for each iteration in Lemma C.2 and Lemma C.3 based on Lemma C.1. Here we are able to choose $\epsilon = O(\frac{\sigma^2}{d^3 L})$ to preserve the condition

$$\mathbb{E}\|\widetilde{g} - \nabla f(x)\|^2 \leq \mathbb{E}\|\widetilde{g} - \nabla f(x)\|_1^2 \leq \sigma^2 \tag{C.7}$$

used in Lemma C.2 and Lemma C.3 with $O(1)$ additional quantum queries. The total number of calls is $O(\frac{T}{h})$ in Lemma C.2 and Lemma C.3, the same scaling as in Lemma B.1 and Lemma B.2. The query complexities of ULD and ULD-RMM are as follows.

**Theorem C.1.** *Assume the target distribution $\rho$ is strongly log-concave with $L$-smooth and $\mu$-strongly convex negative log-density. Let $\rho_n$ be the distribution of the underdamped Langevin diffusion with the initial point $x_0$ satisfying $\|x_0 - x^*\| \leq D$, step size $h \leq \frac{\epsilon}{104\kappa}\sqrt{\frac{1}{d/\mu + D^2}}$, and time $T \geq \frac{\kappa}{2}\log\left(\frac{24\sqrt{d/\mu + D^2}}{\epsilon}\right)$. Then quantum inexact ULD (Algorithm 6) achieves*

$$\mathbb{E}\left(\|\widehat{X}_n - X_T\|^2\right) \leq \widetilde{O}\left(\frac{d^2\kappa^2 h^2}{\mu}\right), \tag{C.8}$$

$$W_2(\rho_n, \rho) \leq \epsilon, \tag{C.9}$$

*using*

$$\frac{T}{h} = \widetilde{\Theta}\left(\frac{\kappa^2\sqrt{d}}{\epsilon}\right) \tag{C.10}$$

*queries to the quantum evaluation oracle.*

*Proof.* By Lemma C.2, we know that the number of iterations of ULD with an inexact gradient oracle is $\widetilde{O}(\kappa^2\sqrt{d}/\epsilon)$, as long as the oracle satisfies $\mathbb{E}\|\widetilde{g} - \nabla f(x)\|^2 \leq \sigma^2$. By Lemma C.1, this condition can be achieved by the quantum gradient algorithm such that each gradient computation takes $O(1)$ queries to the quantum evaluation oracle. Therefore, the total number of queries is $\widetilde{O}(\kappa^2\sqrt{d}/\epsilon)$ for the quantum inexact ULD. □

**Theorem C.2.** *Assume the target distribution $\rho$ is strongly log-concave with $L$-smooth and $\mu$-strongly convex negative log-density. Let $\rho_n$ be the distribution of the randomized midpoint method for underdamped Langevin diffusion with initial point $x_0$, step size $h \leq \min\left\{\frac{\epsilon^{1/3}\mu^{1/6}}{\kappa^{1/6}d^{1/6}\log^{1/6}\left(\frac{\sqrt{d/\mu}}{\epsilon}\right)}, \frac{\epsilon^{2/3}\mu^{1/3}}{d^{1/3}\log^{1/3}\left(\frac{\sqrt{d/\mu}}{\epsilon}\right)}\right\}$, and time $T \geq 2\kappa\log\left(\frac{20d/\mu}{\epsilon^2}\right)$. Then quantum inexact ULD-RMM (Algorithm 7) achieves*

$$\mathbb{E}\left(\|\widehat{X}_n - X_T\|^2\right) \leq \widetilde{O}\left(\frac{d\kappa h^6}{\mu} + \frac{dh^3}{\mu}\right), \tag{C.11}$$

$$W_2(\rho_n, \rho) \leq \epsilon, \tag{C.12}$$

*using*

$$\frac{2T}{h} = \widetilde{\Theta}\left(\frac{\kappa^{7/6}d^{1/6}}{\epsilon^{1/3}} + \frac{\kappa d^{1/3}}{\epsilon^{2/3}}\right) \tag{C.13}$$

*queries to the quantum evaluation oracle.*

The proof is almost the same as Theorem C.1, so we omit it here.

---

**Algorithm 6:** Quantum Inexact Underdamped Langevin Dynamics (Quantum IULD)

---

**Input:** Function $f$, step size $h$, time $T$, and a sample $x_0$ from a starting distribution $\rho_0$, evaluation error $\epsilon$, Lipschitz constant $L$, smoothness parameter $\beta$

**Output:** Sequence $x_h^h, x_{2h}^h, \ldots, x_{\lfloor T \rfloor + 1}^h$

1 Compute $x_0^h \leftarrow x_0$
2 Compute $\widetilde{g}(x_0) \leftarrow \mathsf{SmoothQuantumGradient}(f, \epsilon, L, \beta, x_0)$
3 **for** $t = 0, h, \ldots, \lfloor T \rfloor$ **do**
4 $\quad$ Draw $W_{1,t}^h = \int_0^h e^{2(s-h)}dB_{t+s}$, $W_{2,t}^h = \int_0^h (1 - e^{2(s-h)})dB_{t+s}$
5 $\quad$ Compute $v_{t+h}^h = e^{-2h}v_t^h + \frac{1}{2L}(1 - e^{-2h})\widetilde{g}(x_t^h) + \frac{2}{\sqrt{L}}W_{1,t}^h$
6 $\quad$ $x_{t+h}^h = x_t^h + \frac{1}{2}(1 - e^{-2h})v_t^h + \frac{1}{2L}[h - (1 - e^{-2h})]\widetilde{g}(x_t^h) + \frac{1}{\sqrt{L}}W_{2,t}^h$
7 $\quad$ Compute $\widetilde{g}(x_{t+h}^h) \leftarrow \mathsf{SmoothQuantumGradient}(f, \epsilon, L, \beta, x_{t+h}^h)$

---

**Algorithm 7:** Quantum Inexact Underdamped Langevin Dynamics with Randomized Midpoint Method (Quantum IULD-RMM)

**Input:** Function $f$, step size $h$, time $T$, and a sample $x_0$ from a starting distribution $\rho_0$
**Output:** Sequence $x_h^h, x_{2h}^h, \ldots, x_{\lfloor T \rfloor+1}^h$

1  Compute $x_0^h \leftarrow x_0$, $y_0^h \leftarrow x_0$
2  Compute $\widetilde{g}(x_0^h) \leftarrow \mathsf{SmoothQuantumGradient}(f, \epsilon, L, \beta, x_0^h)$
3  $\widetilde{g}(y_0^h) \leftarrow \mathsf{SmoothQuantumGradient}(f, \epsilon, L, \beta, y_0^h)$
4  **for** $t = 0, h, \ldots, \lfloor T \rfloor$ **do**
5  $\quad$ Draw $W_{1,t}^h = \int_0^h e^{2(s-h)} \mathrm{d}B_{t+s}$, $W_{2,t}^h = \int_0^h (1 - e^{2(s-h)}) \mathrm{d}B_{t+s}$,
$\quad\quad W_{3,t}^h = \int_0^{\alpha h} (1 - e^{2(s-h)}) \mathrm{d}B_{t+s}$
6  $\quad$ Compute $v_{t+h}^h = e^{-2h} v_t^h + \frac{h}{L} e^{-2(1-\alpha)h} \widetilde{g}(y_t^h) + \frac{2}{\sqrt{L}} W_{1,t}^h$
7  $\quad$ $x_{t+h}^h = x_t^h + \frac{1}{2}(1 - e^{-2h}) v_t^h + \frac{h}{2L}(1 - e^{-2(1-\alpha)h}) \widetilde{g}(y_t^h) + \frac{1}{\sqrt{L}} W_{2,t}^h$
8  $\quad$ $y_{t+h}^h = x_t^h + \frac{1}{2}(1 - e^{-2\alpha h}) v_t^h + \frac{1}{2L}[\alpha h - (1 - e^{-2\alpha h})] \widetilde{g}(x_t^h) + \frac{1}{\sqrt{L}} W_{3,t}^h$
9  $\quad$ Compute $\widetilde{g}(x_{t+h}^h) \leftarrow \mathsf{SmoothQuantumGradient}(f, \epsilon, L, \beta, x_{t+h}^h)$
10 $\quad$ $\widetilde{g}(y_{t+h}^h) \leftarrow \mathsf{SmoothQuantumGradient}(f, \epsilon, L, \beta, y_{t+h}^h)$

## C.2 Quantum MALA

In Appendix C.2.1, we introduce several classical results on the mixing of MALA. Then, in Appendix C.2.2, we describe how to implement a quantum walk for MALA. In Appendix C.2.3, we discuss the effective spectral gap of a Markov chain. Then, in Appendix C.2.4 and Appendix C.2.5, we show quantum MALA with a warm start distribution and a Gaussian initial distribution, respectively.

### C.2.1 Mixing time and spectral gap of MALA

The Metropolis adjusted Langevin algorithm (MALA) is a key method for sampling log-concave distributions. Classically, the state-of-the-art mixing time bound of MALA was proven by [28]. They show that MALA is equivalent to the Metropolized Hamiltonian Monte Carlo method (HMC). Then, they consider the following Metropolized HMC algorithm (Algorithm 8) and use the blocking conductance analysis of [25] to upper bound the mixing time.

**Algorithm 8:** Metropolized HMC: HMC$(x_0, \eta)$

**Input:** Initial point $x_0 \in \mathbb{R}^d$, step size $\eta$
**Output:** Sequence $\{x_k\}, k \geq 0$

1  **for** $k \geq 0$ **do**
2  $\quad$ Draw $v_k \sim \mathcal{N}(0, I_d)$
3  $\quad$ $(\widetilde{x}_k, \widetilde{v}_k) \leftarrow \mathsf{LEAPFROG}(\eta, x_k, v_k)$
4  $\quad$ Draw $u \sim \mathcal{U}([0, 1])$
5  $\quad$ **if** $u \leq \min\{1, \exp(\mathcal{H}(x_k, v_k) - \mathcal{H}(\widetilde{x}_k, \widetilde{v}_k))\}$ **then**
6  $\quad\quad$ $x_{k+1} \leftarrow \widetilde{x}_k$
7  $\quad$ **else**
8  $\quad\quad$ $x_{k+1} \leftarrow x_k$

Define $\mathcal{H}(x, v) := f(x) + \frac{1}{2}\|v\|_2^2$. Let $\mathrm{d}\pi^\star$ denote the target distribution, i.e., $\mathrm{d}\pi^\star(x)/\mathrm{d}x \propto \exp(-f(x))$. Then the Markov chain defined by Algorithm 8 has the following property.

**Lemma C.4** ([28]). *The Markov chain of Algorithm 8 is reversible, and its stationary distribution is $\mathrm{d}\pi^\star$.*

The main result of [28] is the following theorem on the mixing time of Algorithm 8.

---
**Algorithm 9:** LEAPFROG($\eta, x, v$)
---
**Input:** Points $x, v \in \mathbb{R}^d$, step size $\eta$
**Output:** Points $\widetilde{x}, \widetilde{v} \in \mathbb{R}^d$
1   $v' \leftarrow v - \frac{\eta}{2}\nabla f(x)$
2   $\widetilde{x} \leftarrow x + \eta v'$
3   $\widetilde{v} \leftarrow v' - \frac{\eta}{2}\nabla f(x)$
---

**Theorem C.3** (Mixing of Hamiltonian Monte Carlo, Theorem 4.7 of [28])**.** *There is an algorithm initialized from a point drawn from $\mathcal{N}(x^\star, L^{-1}I_d)$ that iterates Algorithm 8*

$$O(\kappa d \log(\kappa/\epsilon) \log(d \log(\kappa/\epsilon)) \log(1/\epsilon)) \tag{C.14}$$

*times and produces a point from a distribution $\rho$ such that $\|\rho - \pi^\star\|_{TV} \leq \epsilon$.*

The algorithm in the above theorem defines a new Markov chain where in each step, we draw an integer $j$ uniformly from 0 to $O(\kappa d \log(\kappa/\epsilon) \log(d \log(\kappa/\epsilon)))$ and run Algorithm 8 for $j$ iterations. One step of this Markov chain gives a distribution with TV-distance from $\pi^\star$ at most $(2e)^{-1}$ [28]. Hence, if we run for $\log(1/\epsilon)$ steps, we get $\epsilon$ TV-distance.

Furthermore, we can show that MALA converges faster under a certain warm start condition [11, 15, 47]. We say the initial distribution $\rho_0$ is $\beta$-*warm* if there is a constant $\beta$ independent of $\kappa, d$ such that

$$\sup_{S \in \mathcal{B}(\mathbb{R}^d)} \frac{\rho_n(S)}{\rho(S)} \leq \beta. \tag{C.15}$$

The warmness of the Gaussian $\rho_0 = \mathcal{N}(x^*, \frac{1}{L}I)$ satisfies $\beta \leq \kappa^{d/2}$ [15, 47].

Given a $\beta$-warm initial distribution, MALA has the following improved convergence.

**Lemma C.5** (Theorem 1 of [47])**.** *Assume the target distribution $\rho$ is strongly log-concave with $L$-smooth and $\mu$-strongly convex negative log-density. Let $\rho_n$ be the distribution of the $\frac{1}{2}$-lazy version of MALA with $\beta$-warm initial distribution $\rho_0$ and step size $h = c_0(Ld \log^2(\max\{\kappa, d, \frac{\beta}{\epsilon}, c_2\}))^{-1}$. There exist universal constants $c_0, c_1, c_2 > 0$, such that MALA achieves*

$$d_{TV}(\rho_n, \rho) \leq \epsilon \tag{C.16}$$

*after*

$$n \geq c_1 \kappa \sqrt{d} \log^3(\max\{\kappa, d, \frac{\beta}{\epsilon}, c_2\}) \tag{C.17}$$

*steps.*

### C.2.2   Quantum walks for MALA

The goal of this section is to show quantum speedup for the Metropolis adjusted Langevin algorithm (MALA) using the continuous-space quantum walks defined by [6], which generalize the discrete-time quantum walk of [41] to continuous space.

Given a transition density function $p$, the quantum walk is characterized by the states

$$|\phi_x\rangle := |x\rangle \otimes \int_\Omega \mathrm{d}y \, \sqrt{p_{x \to y}}|y\rangle \qquad \forall x \in \mathbb{R}^n, \tag{C.18}$$

where $p_{x \to y} := p(x, y)$.

Now, denote

$$U := \int_\Omega \mathrm{d}x \, |\phi_x\rangle(\langle x| \otimes \langle 0|), \quad \Pi := \int_\Omega \mathrm{d}x \, |\phi_x\rangle\langle\phi_x|, \quad S := \int_\Omega \int_\Omega \mathrm{d}x \, \mathrm{d}y \, |x, y\rangle\langle y, x|. \tag{C.19}$$

A single step of the quantum walk is defined as the unitary operator

$$W := S(2\Pi - I). \tag{C.20}$$

Alternatively, we can define the quantum walk operator as

$$W' := U^\dagger SUR \cdot U^\dagger SUR,$$

where $R$ is the reflection about the subspace $\mathrm{span}\{|x,0\rangle\}$. It is easy to see that $W$ and $W'$ have the same spectrum. More specifically, if $|u_i\rangle$ is an eigenvector of $W$, then $|v_i\rangle = U^\dagger|u_i\rangle$ is an eigenvector of $W'$ with the same eigenvalue.

The following theorem characterizes the eigenvalues of the quantum walk operator.

**Theorem C.4** (Theorem 3.1 of [6]). *Let*

$$D := \int_\Omega \int_\Omega \mathrm{d}x\,\mathrm{d}y\,\sqrt{p_{x\to y}p_{y\to x}}|x\rangle\langle y| \tag{C.21}$$

*denote the* discriminant operator *of $p$. Let $\Lambda$ be the set of eigenvalues of $D$, so that $D = \int_\Lambda \mathrm{d}\lambda\,\lambda|\lambda\rangle\langle\lambda|$. Then the eigenvalues of the quantum walk operator $W$ in (C.20) are $\pm1$ and $\lambda \pm i\sqrt{1-\lambda^2}$ for all $\lambda \in \Lambda$.*

Furthermore, Ref. [6] shows that, for a reversible Markov chain with unique stationary distribution $\rho$, the state

$$|\rho_W\rangle := \int_\Omega \mathrm{d}x\sqrt{\rho_x}|\phi_x\rangle \tag{C.22}$$

is the unique eigenvalue-1 eigenstate of the quantum walk operator $W$ restricted to the subspace $\mathrm{span}_{\lambda\in\Lambda}\{T|\lambda\rangle, ST|\lambda\rangle\}$. Hence, the stationary distribution corresponds to an eigenstate with eigenphase 0, while the other eigenstates have eigenphase at least $\sqrt{2\delta}$, where $\delta$ is the spectral gap of $P$. Thus, by the quantum phase estimation algorithm with $O(1/\sqrt{\delta})$ calls to $W$, we can distinguish the stationary state from other eigenstates, achieving quadratic speedup over the classical mixing time $O(1/\delta)$.

To implement a quantum version of Algorithm 8, we prepare the initial state $|\pi_0\rangle$ and implement the quantum walk operator $W$.

**Initial state.** For the initial state $|\rho_0\rangle$, by Theorem C.3, it suffices to take $\rho_0 = \mathcal{N}(x^\star, L^{-1}I_d)$, where $x^\star$ is the minimum point of $f(x)$. Suppose we already have $x^\star$. Appendix A.3 of [6] shows that the state

$$\int_{\mathbb{R}^d} \left(\frac{L}{2\pi}\right)^{d/4} e^{-\frac{L}{4}\|z\|_2^2}|z\rangle\mathrm{d}z \tag{C.23}$$

can be efficiently prepared by applying a Box-Muller transformation to the state corresponding to the uniform distribution (i.e., an equal superposition of points). Then, for the $i$th register, we apply the shift operation $U_{\mathrm{shift}}$ with $U_{\mathrm{shift}}|x_i\rangle = |x_i + x_i^\star\rangle$. The resulting state is

$$|\rho_0\rangle = \int_{\mathbb{R}^d} \left(\frac{L}{2\pi}\right)^{d/4} e^{-\frac{L}{4}\|z-x^\star\|_2^2}|z\rangle\mathrm{d}z. \tag{C.24}$$

**Quantum walk operator.** The quantum walk operator $W(P)$ can be implemented[1] using the quantum walk update unitary $U$ that maps each point $|x\rangle$ to the superposition $\int_{\mathbb{R}^d} \mathrm{d}y\sqrt{p_{x\to y}}|y\rangle$. We show how to efficiently implement $U$.

We first use $d$ ancilla registers to prepare a standard Gaussian state

$$|v\rangle := \int_{\mathbb{R}^d} \left(\frac{L}{2\pi}\right)^{d/4} e^{-\frac{L}{4}\|z\|_2^2}|z\rangle\mathrm{d}z. \tag{C.25}$$

Then, we implement a unitary $U_{\mathrm{LF}}$ defined by LeapFrog$(\eta, x, v)$ (Algorithm 9) such that for two points $x, v \in \mathbb{R}^d$, $U_{\mathrm{LF}}|x, v\rangle|0\rangle = |x, v\rangle|\widetilde{x}, \widetilde{v}\rangle$, by querying the gradient oracle of $f$ twice. Then we use another ancilla register to prepare the state

$$|u\rangle := \int_{[0,1]} \mathrm{d}|z\rangle. \tag{C.26}$$

---

[1] As shown in [45], $W(P) = U^\dagger SURU^\dagger SUR$, where $S$ is the swap gate and $R$ is a reflection operator with respect to the state space $\mathrm{span}\{|x\rangle|0\rangle : x \in \mathbb{R}^d\}$.

Based on the registers $x, v, \widetilde{x}, \widetilde{v}, u$, we can decide using two queries to the evaluation oracle for $f$ whether the target register $y$ should be $x$ or $\widetilde{x}$. Overall, this process implements the following mapping (up to a hidden normalization factor):

$$|x\rangle|0\rangle \mapsto |x\rangle \int_{\mathbb{R}^d} \int_{[0,1]} \mathrm{d}v \mathrm{d}u \exp\left(-\|v\|_2^2/4\right)|v\rangle|\widetilde{x}\rangle|\widetilde{v}\rangle|u\rangle|y\rangle. \tag{C.27}$$

Finally, uncomputing the $v, \widetilde{x}, \widetilde{v}$ registers using $U_{\mathrm{LF}}^\dagger$ and the inverse of the unitary preparing $|v\rangle$, we obtain a superposition of points with the correct transition density.

---

**Algorithm 10:** QUANTUMUPDATEUNITARY

**Input:** Quantum state $|\rho\rangle = \int_{\mathbb{R}^d} \mathrm{d}x \sqrt{\rho_x}|x\rangle$.
**Output:** Quantum state $|\phi\rangle = \int_{\mathbb{R}^d} \mathrm{d}x \sqrt{\rho_x}|x\rangle \int_{\mathbb{R}^d} \mathrm{d}y \sqrt{p_{x \to y}}|y\rangle$.
1 Prepare $|\rho\rangle|v\rangle$ where $|v\rangle$ is a $d$-dimensional Gaussian state
2 Apply the leap-frog process to $|\rho\rangle|v\rangle$: $\int_{\mathbb{R}^d} \mathrm{d}x \, \mathrm{d}v|x, v\rangle|\widetilde{x}, \widetilde{v}\rangle$;    /* Query $\mathcal{O}_{\nabla f}$ twice */
3 Prepare $\int_{\mathbb{R}^d} \mathrm{d}x \, \mathrm{d}v|x, v\rangle|\widetilde{x}, \widetilde{v}\rangle \int_{[0,1]} \mathrm{d}u|u\rangle$
4 Compute the target point $y$: $\int_{\mathbb{R}^d} \mathrm{d}x \, \mathrm{d}v \int_{[0,1]} \mathrm{d}u|x, v\rangle|\widetilde{x}, \widetilde{v}\rangle|u\rangle|y\rangle$;    /* Query $\mathcal{O}_f$ twice */
5 Uncompute the $v, \widetilde{x}, \widetilde{v}, u$ registers: $\int \mathrm{d}x \sqrt{\rho_x}|x\rangle \int_{\mathbb{R}^d} \mathrm{d}y \sqrt{p_{x \to y}}|y\rangle$

---

**Algorithm 11:** QUANTUMMALA

**Input:** Evaluation oracle $\mathcal{O}_f$, gradient oracle $\mathcal{O}_{\nabla f}$, initial state $|\rho_0\rangle$
**Output:** Quantum state $|\widetilde{\rho}\rangle$ close to the stationary distribution state $\int_{\mathbb{R}^d} e^{-f(x)} \mathrm{d}|x\rangle$
1 Construct quantum walk update unitary $U$ from QUANTUMUPDATEUNITARY (Algorithm 10) with $\mathcal{O}_f$ and $\mathcal{O}_{\nabla f}$
2 Implement the quantum walk operator $W(P)$
3 Perform $\frac{\pi}{3}$-amplitude amplification with $W(P)$ on the state $|\rho_0\rangle|0\rangle$
4 **return** the resulting state $|\widetilde{\rho}\rangle$

---

**Lemma C.6** (Continuous-space quantum walk implementation). *The Markov chain of Algorithm 8 can be implemented as a continuous-space quantum walk where the quantum walk unitary for one step can be implemented with 2 queries to the gradient oracle, 2 queries to the evaluation oracle, and $O(d)$ quantum gates.*

If we have a sequence of slowly varying log-concave distributions $\rho_0, \ldots, \rho_r$, we can quantumly sample from $\rho_r$ via MALA with a quadratic speedup.

**Theorem C.5** (Quantum speedup for slowly varying Markov chains [45]). *Let $M_0, \ldots, M_r$ be classical reversible Markov chains with stationary distributions $\rho_0, \ldots, \rho_r$ such that each chain has spectral gap at least $\delta^{-1}$. Assume that $|\langle \rho_i | \rho_{i+1} \rangle| \geq p$ for some $p > 0$ and all $i \in \{0, \ldots, r - 1\}$, and that we can prepare the state $|\rho_0\rangle$. Then, for any $\epsilon > 0$, there is a quantum algorithm which produces a quantum state $|\widetilde{\rho}_r\rangle$ such that $\||\widetilde{\rho}_r\rangle - |\rho_r\rangle|0^a\rangle\| \leq \epsilon$, for some integer $a$. The algorithm uses*

$$\widetilde{O}\left(\delta^{-1/2} \cdot \frac{r}{p}\right) \tag{C.28}$$

*applications of the quantum walk operators $W_i'$ corresponding to the chains $M_i$ for $i \in [r]$.*

However, we cannot directly apply Theorem C.5 to speed up MALA, since the spectral gap of MALA is intractable. We only know the mixing time of MALA with some specific initial distributions, instead of any initial distribution.

### C.2.3   Warmness implies effective spectral gap

To overcome the spectral gap issue, we use the idea of the effective spectral gap of a Markov chain. The intuition is that suppose we know the mixing time $t = t_{\mathrm{mix}}(\epsilon, \pi_0)$ for some initial distribution

$\pi_0$, then we can show that $|\pi_0\rangle$ has very small overlap with the eigenspace of $W'$ with the corresponding eigenvalues of $P$ that are very close to $1$. More formally, we prove the following lemma, which improves Proposition 4.2 of [7] that requires the initial distribution to satisfy an $L_2$-norm condition, by instead relying only on the standard warmness of the initial distribution.

**Lemma C.7** (Effective spectral gap for warm start). *Let $M = (\Omega, p)$ be an ergodic reversible Markov chain with a transition operator $P$ and unique stationary state with a corresponding density $\rho$. Let $\{(\lambda_i, f_i)\}$ be the set of eigenvalues and eigenfunctions of $P$, and let $|u_i\rangle$ be the eigenvectors of the corresponding quantum walk operator $W$. Let $\rho_0$ be a probability density that is a warm start for $\rho$ and mixes up to TV-distance $\epsilon$ in $t$ steps of $M$. Furthermore, assume that $\rho_0$ is a $\beta$-warm start of $\rho$. Let $|\phi_{\rho_0}\rangle$ be the state obtained by applying the quantum walk update operator $U$ to the state $|\rho_0\rangle$:*

$$|\phi_{\rho_0}\rangle = \int_\Omega \sqrt{\rho_0(x)} \int_\Omega \sqrt{p_{x\to y}}|x\rangle|y\rangle\, \mathrm{d}x\, \mathrm{d}y. \tag{C.29}$$

*Then $|\langle\phi_{\rho_0}|u_i\rangle| = O(\beta\sqrt{\epsilon})$ for all $i$ with $1 > |\lambda_i| \geq 1 - O(1/t)$.*

*Furthermore, for MALA with $\rho_0$ being $\mathcal{N}(0, L^{-1}I)$, $|\langle\phi_{\rho_0}|u_i\rangle| = O(\sqrt{\epsilon})$.*

**Remark C.1.** *Since $|v_i\rangle = U^\dagger|u_i\rangle$ is the corresponding eigenvector of $W'$, and $|\phi_{\rho_0}\rangle = U|\rho_0, 0\rangle$, Lemma C.7 implies that $|\langle\rho_0, 0|v_i\rangle| \leq \beta\sqrt{\epsilon}$ for any $i$ with $|\lambda_i| \in (1 - O(1/t), 1)$. In other words, effectively, the spectral gap is $\Omega(1/t)$.*

*Proof.* Let $S := \{x \in \mathbb{R}^d : \frac{\rho(x)}{\rho_0(x)} \geq \frac{1}{\epsilon}\}$. Since $\mathbb{E}_{\rho_0}\left[\frac{\rho(x)}{\rho_0(x)}\right] = 1$, by Markov's inequality, we have

$$\int_S \rho_0(x)\mathrm{d}x = \Pr_{\rho_0}[x \in S] \leq \epsilon. \tag{C.30}$$

Then we define a quantum state $|\rho_1\rangle$ such that $\langle\rho_1|x\rangle = \langle\rho_0|x\rangle$ for $x \notin S$, and $\langle\rho_1|x\rangle = 0$ for $x \in S$. Furthermore, let $|\phi_{\rho_1}\rangle := U|\rho_1\rangle$.

We have

$$\||\phi_{\rho_0}\rangle - |\phi_{\rho_1}\rangle\| = \left\|\int_S \sqrt{\rho_0(x)}T|x\rangle\mathrm{d}x\right\| = \left|\int_S \rho_0(x)\mathrm{d}x\right|^{1/2} \leq \sqrt{\epsilon}, \tag{C.31}$$

where $T$ is the isometry

$$T := \int_\Omega \int_\Omega \sqrt{p_{x\to y}}|x, y\rangle\langle x|\, \mathrm{d}x\, \mathrm{d}y.$$

Moreover, by Eqs. (4.35) and (4.36) in [6], if $1 > \lambda_i \geq 1 - O(1/t)$, we have

$$|\langle\phi_{\rho_1}|u_i\rangle| \leq 2|\langle\rho_1|v_i\rangle| = 2\left|\int_{\overline{S}} \sqrt{\frac{\rho(x)}{\rho_0(x)}}\frac{\rho_0(x)f_i(x)}{\rho(x)}\mathrm{d}x\right| \leq \frac{2\langle\rho_0, f_i\rangle_\rho}{\epsilon^{1/2}} = O(\beta\sqrt{\epsilon}), \tag{C.32}$$

where the third step follows from $\frac{\rho(x)}{\rho_0(x)} \leq \frac{1}{\epsilon}$ for $x \notin S$ and the Cauchy-Schwarz inequality, and the last step follows from the claim that $\langle\rho_0, f_i\rangle_\rho = O(\beta\epsilon)$.

Combining these observations, we find

$$|\langle\phi_{\rho_0}|u_i\rangle| \leq |\langle\phi_{\rho_0} - \phi_{\rho_1}|u_i\rangle| + |\langle\phi_{\rho_1}|u_i\rangle| \leq \sqrt{\epsilon} + O(\beta\sqrt{\epsilon}) = O(\beta\sqrt{\epsilon}) \tag{C.33}$$

when $1 > \lambda_i \geq 1 - O(1/t)$, which gives the desired result.

Now, it remains to prove the claim that $\langle\rho_0, f_i\rangle_\rho = O(\beta\epsilon)$ for $1 > \lambda_i > 1 - O(1/t)$. Suppose $\rho_0$ can be decomposed in the eigenbasis of $P$ as $\rho_0 = \rho + \sum_{i=2}^\infty \langle\rho_0, f_i\rangle_\rho f_i$. Then $P^t\rho_0 = \rho + \sum_{i=2}^\infty \lambda_i^t\langle\rho_0, f_i\rangle_\rho f_i$, where $\lambda_i$ is the eigenvalue of $f_i$. Since $\|P^t\rho_0 - \rho\|_1 \leq \epsilon$, by Fact C.1, we have $\|P^t\rho_0 - \rho\|_\rho \leq \beta\epsilon$. Hence, by the orthogonality of $f_i$, we have $\lambda_i^t\langle a, f_i\rangle_\rho \leq \beta\epsilon$. Therefore, when $1 > \lambda_i > 1 - O(1/t)$, we have $\lambda_i^t = \Omega(1)$, which implies that $\langle\rho_0, f_i\rangle_\rho = O(\beta\epsilon)$.

For MALA, by Fact C.1, $\|P^{O(t)}\rho_0 - \rho\|_\rho \leq O(\epsilon)$, which implies that $\langle\rho_0, f_i\rangle_\rho = O(\epsilon)$ by the same argument. Therefore, we get that for MALA with Gaussian initial distribution, $|\langle\phi_{\rho_0}|u_i\rangle| \leq O(\sqrt{\epsilon})$ for $i$ with $1 > |\lambda_i| > 1 - O(1/t)$. $\qquad\square$

**Fact C.1.** *Let $\rho_0$ be a $\beta$-warm start of a Markov chain with transition operator $P$ and stationary distribution $\rho$. If $\|P^t\rho_0 - \rho\|_1 \leq \epsilon$ for some $t > 0$, then $\|P^t\rho_0 - \rho\|_\rho \leq \beta \cdot \epsilon$. For MALA with $\rho_0 = \mathcal{N}(0, L^{-1}I)$, $\|P^{O(t)}\rho_0 - \rho\|_\rho \leq \epsilon$.*

*Proof.* We expand $\|P^t\rho_0 - \rho\|_\rho$ in terms of its definition, giving

$$\|P^t\rho_0 - \rho\|_\rho = \int_{\mathbb{R}^d} \frac{(P^t\rho_0 - \rho)^2(x)}{\rho(x)}\mathrm{d}x = \int_{\mathbb{R}^d} \frac{(P^t\rho_0)^2(x)}{\rho(x)}\mathrm{d}x - 1 = \chi^2(P^t\rho_0, \rho), \qquad \text{(C.34)}$$

where the second step follows since $P^t\rho_0$ and $\rho$ are distributions, and the last step follows from the definition of $\chi^2$-distance. Let $\rho_1 := P^t\rho_0$. By Lemma 27 in [11], we know that $\rho_1$ is also $\beta$-warm. Furthermore, by Lemma 28 in [11], the $\chi$-square distance can be upper bounded by

$$\chi^2(P^t\rho_0, \rho) \leq \beta \cdot \|P^t\rho_0 - \rho\|_1 \leq \beta\epsilon \qquad \text{(C.35)}$$

as claimed. For MALA, as shown in [9], MALA with a Gaussian start also mixes in about $O(t)$ steps in $\sqrt{\chi^2}$ metric. Therefore, we have $\chi^2(P^{O(t)}\rho_0, \rho) \leq \epsilon$. $\square$

### C.2.4 Quantum MALA with a warm start

We first show that quantum MALA can achieve quadratic speedup in query complexity when the initial distribution is a warm start (Theorem C.6).

**Theorem C.6** (Quantum MALA with warm start). *Let $|\rho_0\rangle$ be a $\beta$-warm start with respect to the log-concave distribution $\rho \propto e^{-f}$. Let $t(\epsilon) = \tilde{O}(\kappa\sqrt{d}\log^3(\beta/\epsilon))$ be the mixing time of classical MALA with initial distribution $\rho_0$ as shown in Lemma C.5, i.e., $\|P^{t(\epsilon)}\rho_0 - \rho\|_{\mathrm{TV}} \leq \epsilon$. Then there is a quantum algorithm that prepares a state $|\widetilde{\rho}\rangle$ that is $\epsilon$-close to $|\rho\rangle$ using*

$$\tilde{O}(\kappa^{1/2}d^{1/4}\beta^{1/2}) \qquad \text{(C.36)}$$

*queries to the evaluation oracle $\mathcal{O}_f$ and gradient oracle $\mathcal{O}_{\nabla f}$.*

*Proof.* Let $|\rho_0\rangle$ be the initial state corresponding to a distribution $\rho_0$ that is $\beta$-warm. Then we know that $t = t(\epsilon/\beta)$ steps of the classical MALA random walk suffice to achieve $\|P^t\rho_0 - \rho\|_1 \leq \epsilon/\beta$.

By Lemma C.7, we have $|\rho_0\rangle = |\rho_0'\rangle + |e\rangle$, where $|\rho_0'\rangle$ lies in the space of eigenvectors $|v_i\rangle$ of $W$ such that $\lambda_i = 1$ or $\lambda_i \leq 1 - \tilde{\Omega}(t^{-1})$, and $\||e\rangle\| \leq \epsilon_1$.

Hence, by Corollary 4.1 in [7], the approximate reflection in the quantum walk $\widetilde{R}$ can be implemented using $\widetilde{O}(t^{1/2})$ calls to the controlled-$W$ operator. Furthermore, $\beta$-warmness also implise that $|\langle\rho_0|\rho\rangle| \geq \beta^{-1/2}$. Thus, the approximated stationary state $|\widetilde{\rho}\rangle$ can be prepared via $O(\beta^{1/2}\log(1/\epsilon_2))$ recursive levels of $\frac{\pi}{3}$-amplitude amplification such that $\||\rho\rangle - |\widetilde{\rho}\rangle\| \leq \epsilon_2$.

By choosing $\epsilon_1 = \epsilon/(2\log(2/\epsilon))$ and $\epsilon_2 = \epsilon/2$, we achieve a final approximation error of $O(\epsilon_1\log(\epsilon_2) + \epsilon_2) \leq \epsilon$. By Lemma C.6, each controlled-$W$ operator takes a constant number of queries to $\mathcal{O}_f$ and $\mathcal{O}_{\nabla f}$. Therefore, by plugging-in the classical mixing time of MALA, we obtain the desired query complexity. $\square$

---

**Algorithm 12:** QUANTUMMALAWITHWARMSTART

---

**Input:** Evaluation oracle $\mathcal{O}_f$, gradient oracle $\mathcal{O}_{\nabla f}$, smoothness parameter $L$, convexity parameter $\mu$, warm-start state $|\rho_0\rangle$

**Output:** Quantum state $|\widetilde{\rho}\rangle$ close to the stationary distribution state $\int_{\mathbb{R}^d} e^{-f(x)}\mathrm{d}|x\rangle$

1 $|\widetilde{\rho}\rangle \leftarrow$ QUANTUMMALA$(\mathcal{O}_{f_i}, \mathcal{O}_{\nabla f_i}, |\rho_0\rangle)$ (Algorithm 11)
2 **return** $|\widetilde{\rho}\rangle$

---

### C.2.5 Quantum MALA with Gaussian-start

We cannot directly apply Theorem C.6 for a Gaussian initial distribution because the overlap between $|\rho_0\rangle$ and $|\rho\rangle$ is exponentially small. Instead, we use the idea of simulated annealing and construct a sequence of slowly-varying Markov-chains (as in [45]). We have the following result, which looks like Theorem C.5. But our result uses the effective spectral gap of MALA (by Lemma C.7).

**Corollary C.1** (Quantum speedup for slowly varying MALAs)**.** *Let $\rho_0, \ldots, \rho_r$ be a sequence of log-concave distributions such that $|\langle\rho_i|\rho_{i+1}\rangle| \geq p$ for some $p > 0$ and for all $i \in \{0, \ldots, r-1\}$. Suppose we can prepare the initial state $|\rho_0\rangle$. Then, for any $\epsilon > 0$, there is a quantum procedure to produce a state $|\widetilde{\rho}_r\rangle$ such that $\||\widetilde{\rho}_r\rangle - |\rho_r\rangle\| \leq \epsilon$ using*

$$\widetilde{O}(\kappa^{1/2}d^{1/2} \cdot (r/p)) \tag{C.37}$$

*applications of the quantum walk operators $W_i$ corresponding to the MALA procedure for $\rho_i$ for $i \in [r]$.*

*Proof.* For two consecutive distributions, quantum MALA can evolve $|\rho_i\rangle$ to $|\rho_{i+1}\rangle$ using $\widetilde{O}(\sqrt{\kappa d} \cdot p^{-1})$ quereis, which follows from Theorem C.6 and Theorem C.3 (which holds for all initial distributions with warmness $\beta \leq \kappa^{d/2}$). Then, the corollary immediately follows. $\qquad\square$

---

**Algorithm 13:** QUANTUMMALAFORLOG-CONCAVESAMPLING

**Input:** Evaluation oracle $\mathcal{O}_f$, gradient oracle $\mathcal{O}_{\nabla f}$, smoothness parameter $L$, convexity parameter $\mu$

**Output:** Quantum state $|\widetilde{\rho}\rangle$ close to the stationary distribution state $\int_{\mathbb{R}^d} e^{-f(x)}\mathrm{d}|x\rangle$

1 Compute the cooling schedule parameters $\sigma_1, \ldots, \sigma_M$
2 Prepare the state $|\rho_0\rangle \propto \int_{\mathbb{R}^d} e^{-\frac{1}{4}\|x\|^2/\sigma_1^2}\mathrm{d}|x\rangle$
3 **for** $i \leftarrow 1, \ldots, M$ **do**
4 $\quad$ Construct $\mathcal{O}_{f_i}$ and $\mathcal{O}_{\nabla f_i}$ where $f_i(x) = f(x) + \frac{1}{2}\|x\|^2/\sigma_i^2$
5 $\quad$ $|\rho_i\rangle \leftarrow$ QUANTUMMALA$(\mathcal{O}_{f_i}, \mathcal{O}_{\nabla f_i}, |\rho_{i-1}\rangle)$ (Algorithm 11)
6 **return** $|\rho_M\rangle$

---

**Theorem C.7** (Quantum MALA for log-concave sampling)**.** *Assume the target distribution $\rho \propto e^{-f}$ is strongly log-concave with $f : \mathbb{R}^d \to \mathbb{R}_+$ being L-smooth and $\mu$- strongly convex. Let $|\rho\rangle$ be the quantum state corresponding to the distribution $\rho$. Then, for any $\epsilon > 0$, there is a quantum algorithm (Algorithm 13) that prepares a state $|\widetilde{\rho}\rangle$ such that $\||\widetilde{\rho}\rangle - |\rho\rangle\| \leq \epsilon$ using $\widetilde{O}(\kappa^{1/2}d)$ queries to the evaluation oracle $\mathcal{O}_f$ and gradient oracle $\mathcal{O}_{\nabla f}$.*

*Proof.* By Lemma B.6, we know that the cooling schedule $\sigma_1, \ldots, \sigma_M$ gives a sequence of slowly-varying Markov chains with overlap $p = \Omega(1)$. We also know that the length of the schedule is $M = \widetilde{O}(d^{1/2})$.

Hence, by Corollary C.1 with $r = \widetilde{O}(d^{1/2})$ and $p = \Omega(1)$, $|\widetilde{\rho}\rangle$ can be prepared by $\widetilde{O}(\kappa^{1/2}d)$ quantum walk steps. By Lemma C.6, each step queries $\mathcal{O}_f$ and $\mathcal{O}_{\nabla f}$ twice. The result follows. $\qquad\square$

## D Quantum Algorithm for Estimating Normalizing Constants: Details

We now come back to the problem of estimating the normalizing constant.

### D.1 Quantum MALA and annealing

In this section, we first describe a quantum speedup for the annealing process via a quantum-accelerated Monte Carlo method, which quadratically improves the $\epsilon$-dependence of the sampling complexity of the classical algorithm. Then we further reduce the $\kappa$- and $d$-dependence of the query complexity using the quantum MALA procedure developed in Appendix C.2.

### D.1.1 Quantum speedup for the standard annealing process

Reference [31] developed a quantum-accelerated Monte Carlo method for mean estimation with $B$-bounded relative variance.[2] We state the result as follows.

**Lemma D.1** (Theorem 6 of [31]). *Assume there is an algorithm $\mathcal{A}$ such that $v(\mathcal{A}) \geq 0$ and $\frac{\mathrm{Var}(v(\mathcal{A}))}{\mathbb{E}[v(\mathcal{A})]^2} \leq B$ for some $B \geq 1$, and an accuracy $\epsilon < 27B/4$. Then there is a quantum algorithm which outputs an estimate $\widetilde{\mu}$ such that*

$$\Pr\left[ |\widetilde{\mu} - \mathbb{E}[v(\mathcal{A})]| \geq \epsilon \mathbb{E}[v(\mathcal{A})] \right] \leq \frac{1}{4}, \tag{D.1}$$

*with*

$$O\left( \frac{B}{\epsilon} \log^{3/2}\left( \frac{B}{\epsilon} \right) \log\log\left( \frac{B}{\epsilon} \right) \right) \tag{D.2}$$

*queries to $\mathcal{A}$.*

**Lemma D.2** (Lemma 1 of [31]). *Let $\mathcal{A}$ be a (classical or quantum) algorithm which aims to estimate some quantity $\mu$, and whose output $\tilde{\mu}$ satisfies $|\mu - \tilde{\mu}| \leq \epsilon$ except with probability $\gamma$, for some fixed $\gamma < 1/2$. Then, for any $\delta > 0$, it suffices to repeat $\mathcal{A}$ $O(\log 1/\delta)$ times and take the median to obtain an estimate which is accurate to within $\epsilon$ with probability at least $1 - \delta$.*

This result provides a way of estimating the telescoping product (B.25). The following theorem and its proof closely follows Theorem 8 of [31], while the definitions of the partition function and the cooling schedule are different.

**Theorem D.1** (Quantum speedup of annealing). *Let $Z$ be the normalizing constant in (1.3). Consider a sequence of values $g_i$ as in (B.28), with $\frac{\mathbb{E}_{\rho_i}(g_i^2)}{\mathbb{E}_{\rho_i}(g_i)^2} = O(1)$. Further assume that we have the ability to sample $\rho_i$ for $i \in [M]$. Then there is a quantum algorithm which outputs an estimate $\widetilde{Z}$, such that*

$$\Pr[(1 - \epsilon)Z \leq \widetilde{Z} \leq (1 + \epsilon)Z] \geq \frac{3}{4}, \tag{D.3}$$

*using*

$$O\left( \frac{M^2}{\epsilon} \log^{3/2}\left( \frac{M}{\epsilon} \right) \log\log\left( \frac{M}{\epsilon} \right) \right) = \widetilde{O}\left( \frac{M^2}{\epsilon} \right) \tag{D.4}$$

*samples in total.*

*Proof.* For $i \in [M]$, we estimate $\mathbb{E}_{\rho_i}(g_i)$ with output $\widetilde{g}_i$ up to additive error $(\epsilon/2M)\mathbb{E}_{\rho_i}(g_i)$ with failure probability $1/4M$. We output as a final estimate

$$\widetilde{Z} = \widetilde{Z}_1 \prod_{i=1}^{M} \widetilde{g}_i, \tag{D.5}$$

where $\widetilde{Z}_1$ is the normalizing constant of the Gaussian distribution with variance $\sigma_1^2$ as in Lemma B.3. Assuming that all the estimates are indeed accurate, we have

$$1 - \epsilon \leq \left(1 - \frac{\epsilon}{2}\right)\left(1 - \frac{\epsilon}{2M}\right)^M \leq \frac{\widetilde{Z}}{Z} \leq \left(1 + \frac{\epsilon}{2M}\right)^M \leq e^{\epsilon/2} \leq 1 + \epsilon. \tag{D.6}$$

Thus $|\widetilde{Z} - Z| \leq \epsilon Z$ with probability at least

$$\left(1 - \frac{1}{4M}\right)^M \geq 1 - \frac{1}{4} = \frac{3}{4}. \tag{D.7}$$

Based on Lemma B.4 and Lemma B.5, $\frac{\mathbb{E}_{\rho_i}(g_i^2)}{\mathbb{E}_{\rho_i}(g_i)^2} = O(1)$, so $\frac{\mathrm{Var}_{\rho_i}(g_i)}{\mathbb{E}_{\rho_i}(g_i)^2} = O(1)$. By Lemma D.1, each requires

$$O\left( \frac{M}{\epsilon} \log^{3/2}\left( \frac{M}{\epsilon} \right) \log\log\left( \frac{M}{\epsilon} \right) \right) \tag{D.8}$$

---

[2]Reference [4] improves the scaling from $\widetilde{O}(B/\epsilon)$ to $\widetilde{O}(\sqrt{B}/\epsilon)$. Such a result also follows from the quantum Chebyshev inequality of [21]. Since $B = O(1)$ in our case, we apply the original algorithm of [31].

samples from $\rho_i$, and the total number of samples is

$$O\left(\frac{M^2}{\epsilon} \log^{3/2}\left(\frac{M}{\epsilon}\right) \log\log\left(\frac{M}{\epsilon}\right)\right). \tag{D.9}$$

This completes the proof. $\qquad\square$

### D.1.2 Quantum MALA for estimating the normalizing constant

We now describe how to combine quantum annealing with quantum MALA to reduce the query complexity of estimating the normalizing constants.

We begin with the following lemma on non-destructive mean estimation.

**Lemma D.3** (Non-destructive mean estimation with quantum MALA). *For $\epsilon < 1$, given $\widetilde{O}(\epsilon^{-1})$ copies of a state $|\widetilde{\rho}_{i-1}\rangle$ such that $\||\widetilde{\rho}_{i-1}\rangle - |\rho_{i-1}\rangle\| \le \epsilon$, there exists a quantum procedure that outputs $\widetilde{g}_i$ such that*

$$|\widetilde{g}_i - \mathbb{E}_{\rho_i}[g_i]| \le \epsilon \cdot \mathbb{E}_{\rho_i}[g_i] \tag{D.10}$$

*with success probability $1 - o(1)$ using*

$$\widetilde{O}(\kappa^{1/2} d^{1/2} \epsilon^{-1}) \tag{D.11}$$

*steps of the quantum walk operator corresponding to the MALA with stationary distribution $\rho_i$, where $\delta$ is the spectral gap of the Markov chain. The quantum procedure also returns $\widetilde{O}(\epsilon^{-1})$ copies of the state $|\widetilde{\rho}_i\rangle$ such that $\||\widetilde{\rho}_i\rangle - |\rho_i\rangle\| \le \epsilon$.*

*Proof sketch.* This lemma is a variant of Lemma 4.4 in [7]. Notice that Corollary C.1 implies that we can prepare $|\widetilde{\rho}_i\rangle$ from $|\widetilde{\rho}_{i-1}\rangle$ using $\widetilde{O}(\kappa^{1/2} d^{1/2} p^{-1})$ quantum walk steps, where $p \le |\langle \rho_i | \rho_{i-1}\rangle|$. By Lemma B.6, we have $p = \Omega(1)$. The lemma follows by properly choosing the parameters in Lemma 4.4 in [7]. $\qquad\square$

**Theorem D.2** (Quantum speedup using MALA, annealing, and quantum walk). *Let $Z$ be the normalizing constant in (1.3). Assume we are given the access to query the quantum gradient oracle (1.5). Then there is a quantum algorithm which outputs an estimate $\widetilde{Z}$, such that*

$$\Pr[(1-\epsilon)Z \le \widetilde{Z} \le (1+\epsilon)Z] \ge \frac{3}{4}, \tag{D.12}$$

*using*

$$\widetilde{O}\left(d^{3/2} \kappa^{1/2} \epsilon^{-1}\right) \tag{D.13}$$

*queries to the quantum gradient oracle in total.*

*Proof.* The number of annealing stages is $M = \widetilde{O}(\sqrt{d})$. At the $i$th stage, we estimate $\mathbb{E}_{\rho_i}[g_i]$ with relative error $\epsilon/M$. Hence, we can apply Lemma D.3 $M$ times, where each application takes

$$\widetilde{O}(\kappa^{1/2} d^{1/2} (\epsilon/M)^{-1}) = \widetilde{O}(\kappa^{1/2} d\epsilon^{-1}) \tag{D.14}$$

MALA quantum walk steps. This process takes

$$M \cdot \widetilde{O}(\kappa^{1/2} d\epsilon^{-1}) = \widetilde{O}(\kappa^{1/2} d^{3/2} \epsilon^{-1}) \tag{D.15}$$

steps in total.

By Lemma C.6, each step of the quantum walk operator can be implemented by querying the gradient oracle and the evaluation oracle $O(1)$ times. Therefore, we can estimate $Z$ with relative error $\epsilon$ using $\widetilde{O}(\kappa^{1/2} d^{3/2} \epsilon^{-1})$ queries to the gradient and evaluation oracles. $\qquad\square$

## D.2 Quantum multilevel Langevin algorithms

We now consider an alternative approach for estimating the normalizing constant, by replacing MALA by a multilevel Langevin approach. More concretely, for each sample we perform the underdamped Langevin diffusion (ULD) or the randomized midpoint method for underdamped Langevin diffusion (ULD-RMM) that has an improved dependence on the dimension, and apply the multilevel Monte Carlo (MLMC) to preserve the dependence on the accuracy.

Multilevel Monte Carlo methods have attracted extensive attention in stochastic simulations and financial models [17, 18]. This approach was originally developed by [23] for parametric integration, and used to simulate SDEs in [17]. Considering a general random variable $P$, MLMC gives a sequence of estimators $P_0, P_l, \ldots, P_L$ for approximating $P$ with increasing accuracy and cost, and uses the telescoping sum of $\mathbb{E}[P_l - P_{l-1}]$ to estimate $\mathbb{E}[P]$. For $P_l - P_{l-1}$ with smaller variance but larger cost, MLMC performs fewer samples to reach a given error tolerance, reducing the overall complexity. MLMC has been widely discussed and improved under many settings, and has been used in various applications [18].

To estimate normalizing constants, a variant of MLMC has been proposed by [16, Lemmas C.1 and C.2]. Unlike standard MLMC for bounding the mean-squared error, this approach upper bounds the bias and the variance separately, making the analysis more technically involved. The first quantum algorithm based on MLMC was developed by [2, Theorem 2]. They upper bound the additive error with high probability (as is common for quantum algorithms). They also observe that the mean-squared error can control both the bias and the variance [2, Section 2.2] and that the mean-squared error is almost equivalent to the additive error with high probability [2, Appendix A]. Considering this, we still use the additive error scenario for estimating normalizing constants, both for convenience and for compatibility with the quantum annealing speedup of Theorem D.1.

We first introduce the general quantum speedup of MLMC as described in [2], and then apply these results to our problem.

### D.2.1 Quantum-accelerated multilevel Monte Carlo method

We begin by describing the following general result on quantum-accelerated multilevel Monte Carlo (QA-MLMC).

**Lemma D.4** (Theorem 2 of [2]). *Let $P$ denote a random variable, and let $P_l$ (for $l \in \{0, 1, \ldots, L\}$) denote a sequence of random variables such that $P_l$ approximates $P$ at level $l$. Also define $P_{-1} = 0$. Let $C_l$ be the cost of sampling from $P_l$, and let $V_l$ be the variance of $P_l - P_{l-1}$. If there exist positive constants $\alpha, \beta = 2\hat{\beta}, \gamma$ such that $\alpha \geq \min(\hat{\beta}, \gamma)$ and*

- $|\mathbb{E}[P_l - P]| = O(2^{-\alpha l})$,
- $V_l = O(2^{-\beta l}) = O(2^{-2\hat{\beta}l})$, *and*
- $C_l = O(2^{\gamma l})$,

*then for any $\epsilon < 1/e$ there is a quantum algorithm that estimates $\mathbb{E}[P]$ up to additive error $\epsilon$ with probability at least 0.99, and with cost*

$$\begin{cases} O\left(\epsilon^{-1}(\log 1/\epsilon)^{3/2}(\log \log 1/\epsilon)^2\right), & \hat{\beta} > \gamma, \\ O\left(\epsilon^{-1}(\log 1/\epsilon)^{7/2}(\log \log 1/\epsilon)^2\right), & \hat{\beta} = \gamma, \\ O\left(\epsilon^{-1-(\gamma-\hat{\beta})/\alpha}(\log 1/\epsilon)^{3/2}(\log \log 1/\epsilon)^2\right), & \hat{\beta} < \gamma. \end{cases} \tag{D.16}$$

We apply this result to the payoff model of general stochastic differential equations (SDEs) as discussed in [2]. Consider an SDE

$$\mathrm{d}X_t = \mu(X_t, t)\,\mathrm{d}t + \sigma(X_t, t)\,\mathrm{d}W_t \tag{D.17}$$

for $t \in [0, T]$, where we assume $\mu$ and $\sigma$ are Lipschitz continuous. Given an initial condition $X_0$ and an evolution time $T > 0$, we aim to compute

$$\mathbb{E}[\mathcal{P}(X_T)], \tag{D.18}$$

where $\mathcal{P}(X)$ is the so-called payoff function as a functional of $X$. In Lemma D.4, we denote $\mathcal{P}(X_T)$ as $P$, and the goal is to estimate $\mathbb{E}[P]$.

We also consider a numerical scheme that produces $\widehat{X}_k$ with time step size $h = T/n$. We say the scheme is of strong order $r$ if for any $m \in \{1, 2\}$, there exists a constant $C_m$ such that

$$\mathbb{E}\left(\|\widehat{X}_n - X_T\|^m\right) \le C_m h^{rm}. \tag{D.19}$$

Note that it suffices to verify this condition for $m = 2$ since $(\mathbb{E}\|\widehat{X}_n - X_T\|)^2 \le \mathbb{E}\|\widehat{X}_n - X_T\|^2$. We further assume the coefficients of the scheme are Lipschitz continuous. For the discretization $n_l = 2^l$ with step size $h = T/2^l$, we let $P_l$ denote $\mathcal{P}(\widehat{X}_{n_l})$, an estimator of $P$.

Finally, we assume $\mathcal{P}(X)$ is $L_P$-Lipschitz continuous. Thus, we have satisfied the three assumptions of Proposition 2 of [2], which estimates the rates of $|\mathbb{E}[P_l - P]|, V_l, C_l$ in Lemma D.4. We state a simpler version as follows.

**Lemma D.5** (Proposition 2 of [2]). *Given an SDE and a scheme of strong order $r$ with Lipschitz continuous constants, and given a Lipschitz continuous payoff function $\mathcal{P}$, we have $\alpha = r$, $\beta = 2r$, and $\gamma = 1$.*

Note that while we relax the definition of a scheme of strong order $r$, our definition (D.19) is sufficient to prove Lemma D.5. More concretely, in the proof of Proposition 2 of [2], we have the following simplified inequalities:

$$|\mathbb{E}[P_l - P]| \le \mathbb{E}|\mathcal{P}(\widehat{X}_n) - \mathcal{P}(X_T)| \le L_P \mathbb{E}\|\widehat{X}_n - X_T\| \le L_P C_1 h^r = O(2^{-rl}), \tag{D.20}$$

$$V_l \le \mathbb{E}|\mathcal{P}(\widehat{X}_n) - \mathcal{P}(X_T)|^2 \le L_P \mathbb{E}\|\widehat{X}_n - X_T\|^2 \le L_P C_2 h^{2r} = O(2^{-2rl}), \tag{D.21}$$

$$C_l = O(n_l) = O(2^l), \tag{D.22}$$

and therefore $\alpha = r$, $\beta = 2r$, and $\gamma = 1$.

Finally, we can characterize the performance of QA-MLMC as follows.

**Lemma D.6** (Theorem 3 of [2]). *Given an SDE and a scheme of strong order $r$ with Lipschitz continuous constants, and given a Lipschitz continuous payoff function $\mathcal{P}$, QA-MLMC estimates $\mathbb{E}[\mathcal{P}]$ up to additive error $\epsilon$ with probability at least 0.99 with cost*

$$O\left(\epsilon^{-1}(\log 1/\epsilon)^{3/2}(\log \log 1/\epsilon)^2\right), \quad r > 1, \tag{D.23}$$

$$O\left(\epsilon^{-1}(\log 1/\epsilon)^{7/2}(\log \log 1/\epsilon)^2\right), \quad r = 1, \tag{D.24}$$

$$O\left(\epsilon^{-1/r}(\log 1/\epsilon)^{3/2}(\log \log 1/\epsilon)^2\right), \quad r < 1. \tag{D.25}$$

Note that we can amplify the success probability to $1 - \delta$ for arbitrarily small $\delta > 0$ using the powering lemma (Lemma D.2).

### D.2.2 Quantum-accelerated multilevel Langevin method

We have described ULD and ULD-RMM in Algorithm 4 and Algorithm 5, respectively. We now apply these schemes to simulate the underdamped Langevin dynamics as the SDE. According to (D.19), ULD and ULD-RMM are schemes of order 1 and 1.5, respectively.

Let the payoff function $\mathcal{P}$ be $g_i$ as defined in (B.28). Our goal is to estimate the mean of $\mathcal{P}(\widehat{X}_{n_l}) = g_i(\widehat{X}_{n_l})$ using several samples $\widehat{X}_{n_l}$ produced by ULD or ULD-RMM. If $g$ is assumed to be $L_g$-Lipschitz as in Lemma C.2 of [16], we have a Lipschitz continuous payoff function $\mathcal{P}$ with $L_P = L_g$. Although $g_i = \exp\left(\frac{\|x\|^2}{\sigma_i^2(1+\alpha^{-1})}\right)$ is not Lipschitz, according to Section 4.3 of [16], we can truncate at large $x$ and replace $g_i$ by $h_i := \min\left\{g_i, \exp\left(\frac{(r_i^+)^2}{\sigma_i^2(1+\alpha^{-1})}\right)\right\}$ with

$$\alpha = \widetilde{O}\left(\frac{1}{\sqrt{d}\log(1/\epsilon)}\right), \tag{D.26}$$

$$r_i^+ = \mathbb{E}_{\rho_{i+1}}\|x\| + \Theta(\sigma_i\sqrt{(1+\alpha)\log(1/\epsilon)}), \tag{D.27}$$

to ensure $\frac{h_i}{\mathbb{E}_{\rho_i} g_i}$ is $O(\sigma_i^{-1})$ Lipschitz and $\left|\mathbb{E}_{\rho_i}(h_i - g_i)\right| < \epsilon$ by Lemmas C.7 and C.8 of [16]. For simplicity, as in Section 4.2 of [16], we regard $g_i$ as a Lipschitz continuous payoff function in our main results.

Thus, using Lemma D.6 to estimate $\frac{Z_{i+1}}{Z_i} = \mathbb{E}_{\rho_i}(g_i)$, QA-MLMC using either ULD or ULD-RMM can reduce the $\epsilon$-dependence of the number of steps to $\widetilde{O}(\epsilon^{-1})$. Then each step of ULD or ULD-RMM uses the value of $\nabla f(x)$ about $O(1)$ times as shown in Algorithm 4 and Algorithm 5.

Having described the implementations of quantum inexact ULD and ULD-RMM, we now state the quantum speedup for estimating normalizing constants using multilevel ULD and annealing, or multilevel ULD-RMM and annealing, as follows.

**Theorem D.3** (Quantum speedup using multilevel ULD and annealing)**.** *Let $Z$ be the normalizing constant in (1.3). Assume we are given access to the quantum gradient oracle (1.5). Then there is a quantum algorithm which outputs an estimate $\widetilde{Z}$ such that*

$$\Pr[(1 - \epsilon)Z \le \widetilde{Z} \le (1 + \epsilon)Z] \ge \frac{3}{4} \tag{D.28}$$

*using*

$$\widetilde{O}\Big(\frac{d^{3/2}\kappa^2}{\epsilon}\Big) \tag{D.29}$$

*queries to the quantum gradient oracle.*

*Proof.* As in Theorem D.1 and Theorem D.2, for $i \in [M]$ with $M$ stages, we estimate $\mathbb{E}_{\rho_i}(g_i)$ with output $\widetilde{g}_i$ up to additive error $(\epsilon/2M)\mathbb{E}_{\rho_i}(g_i)$ with failure probability $1/4M$, which ensures $|\widetilde{Z} - Z| \le \epsilon Z$ with probability at least $\frac{3}{4}$.

According to Lemma D.6, each sample of $\rho_i$ using multilevel ULD uses $\widetilde{O}(\frac{M\kappa^2\sqrt{d}}{\epsilon})$ queries to the quantum evaluation oracle (1.4) used in Algorithm 6 or Algorithm 7. For $M = \widetilde{O}(\sqrt{d})$ stages, the query complexity of estimating the normalizing constant is $\widetilde{O}(\frac{M^2\kappa^2\sqrt{d}}{\epsilon}) = \widetilde{O}(\frac{d^{3/2}\kappa^2}{\epsilon})$. $\qquad\square$

**Theorem D.4** (Quantum speedup using multilevel ULD-RMM and annealing)**.** *Let $Z$ be the normalizing constant in (1.3). Assume we are given the access to the quantum gradient oracle (1.5). Then there is a quantum algorithm which outputs an estimate $\widetilde{Z}$ such that*

$$\Pr[(1 - \epsilon)Z \le \widetilde{Z} \le (1 + \epsilon)Z] \ge \frac{3}{4} \tag{D.30}$$

*using*

$$\widetilde{O}\Big(\frac{d^{7/6}\kappa^{7/6} + d^{4/3}\kappa}{\epsilon}\Big) \tag{D.31}$$

*queries to the quantum gradient oracle.*

*Proof.* As above, for $i \in [M]$ with $M$ stages, we estimate $\mathbb{E}_{\rho_i}(g_i)$ with output $\widetilde{g}_i$ up to additive error $(\epsilon/2M)\mathbb{E}_{\rho_i}(g_i)$ with failure probability $1/4M$.

According to Lemma D.6 and Lemma B.2, each sample of $\rho_i$ using multilevel ULD uses $\widetilde{O}(\frac{M(\kappa^{7/6}d^{1/6}+\kappa d^{1/3})}{\epsilon})$ queries to the quantum gradient oracle (1.5) or the quantum evaluation oracle (1.4) (with additional $\widetilde{O}(1)$ cost). For $M = \widetilde{O}(\sqrt{d})$ stages, the query complexity of estimating the normalizing constant is $\widetilde{O}(\frac{M^2(\kappa^{7/6}d^{1/6}+\kappa d^{1/3})}{\epsilon}) = \widetilde{O}(\frac{d^{7/6}\kappa^{7/6}+d^{4/3}\kappa}{\epsilon})$. $\qquad\square$

# E   Proof of the Quantum Lower Bound

To prove Theorem 5.1, we use the following quantum query lower bound on the Hamming weight problem.

**Proposition E.1** (Theorem 1.3 of [32]). *For $x = (x_1, \ldots, x_n) \in \{0,1\}^n$, let $\|x\|_1 = \sum_{i=1}^n x_i$ be the Hamming weight of $x$. Furthermore, let $\ell, \ell'$ be integers such that $0 \le \ell < \ell' \le n$. Define the partial boolean function $f_{\ell,\ell'}$ on $\{0,1\}^n$ as*

$$f_{\ell,\ell'}(x) = \begin{cases} 0 & \text{if } \|x\|_1 = \ell \\ 1 & \text{if } \|x\|_1 = \ell'. \end{cases} \tag{E.1}$$

*Let $m \in \{\ell, \ell'\}$ be such that $|\frac{n}{2} - m|$ is maximized, and let $\Delta = \ell' - \ell$. Then given the quantum query oracle*

$$O_x|i\rangle|b\rangle = |i, b \oplus x_i\rangle \quad \forall i \in [n], b \in \{0,1\}, \tag{E.2}$$

*the quantum query complexity of computing the function $f_{\ell,\ell'}$ is $\Theta(\sqrt{n/\Delta} + \sqrt{m(n-m)}/\Delta)$.*

Now we prove Theorem 5.1 using a construction motivated by Section 5 of [16].

*Proof.* We start from a basic function $f_0(x) = \frac{\|x\|^2}{2}$. The partition function of $f_0$ is

$$\int_{\mathbb{R}^k} e^{-f_0(x)} \mathrm{d}x = (2\pi)^{k/2}. \tag{E.3}$$

We then construct $n$ cells in $\mathbb{R}^d$. Without loss of generality we assume that $n^{1/k}$ is an integer, and let $l := 1/(\sqrt{k}n^{1/k})$. We partition $[-1/\sqrt{k}, 1/\sqrt{k}]$ into $n^{1/k}$ intervals, each having length $2l$. Let $I_i$ denote the $i^{\text{th}}$ interval, where $i \in [n^{1/k}]$. Each cell will thus be represented as a $k$-tuple $(i_1, \ldots, i_k) \in \{1, 2, \ldots, n^{1/k}\}^k$ corresponding to $I_{i_1} \times \cdots \times I_{i_k} \subset \mathbb{R}^k$.

Next, each cell $\tau = (i_1, \ldots, i_k)$ with center denoted $v_\tau$ is assigned one of two types (as detailed below), and we let

$$f(x) = \begin{cases} f_0(x) & \text{if cell } \tau \text{ is of type 1} \\ f_0(x) + c_\tau q(\frac{1}{l}(x - v_\tau)) & \text{if cell } \tau \text{ is of type 2}. \end{cases} \tag{E.4}$$

The function $q$ and the normalizing constant $c_\tau$ are carefully chosen, following Lemma D.1 in [16], such that

- $f(x)$ is 1.5-smooth and 0.5-strongly convex; and

- the partition function $Z_f = \int_{\mathbb{R}^k} e^{-f(x)} \mathrm{d}x = (2\pi)^{k/2} - C \cdot \frac{n_2}{n}$, where $n_2$ is the number of type-2 cells, and $C$ is at least $\Omega(l^2)$.

With these properties, we consider two functions as follows. We choose $\delta$ such that $\epsilon = \Theta(\delta^{1+4/k})$. One of the functions has a $1/2 + \delta$ fraction of its cells of type 1 (and a $1/2 - \delta$ fraction of type 2). The other function has a $1/2 - \delta$ fraction of its cells of type 1 (and a $1/2 + \delta$ fraction of type 2). Note that one query to the quantum evaluation oracle (1.4) can be implemented using one quantum query to the binary information indicating the type of the corresponding cell. In addition, by Proposition E.1 with $\ell = (1 - \delta)n/2$ and $\ell' = (1 + \delta)n/2$, it takes $\Omega(1/\delta)$ quantum queries to distinguish whether there are $(1 + \delta)n/2$ or $(1 - \delta)n/2$ cells of type 1. Since $C = \Omega(l^2)$, the partition functions of the two functions differ by a multiplicative factor of at least $1 + \Omega(l^2\delta)$, where $l = \Theta(n^{-1/k}) = \Theta(\delta^{2/k})$, and hence $l^2\delta = \Theta(\delta^{1+4/k}) = \Theta(\epsilon)$. The quantum query complexity is therefore $\Omega(1/\delta) = \Omega(\epsilon^{-\frac{1}{1+4/k}})$ as claimed. $\square$