# OpenReview forum: "Quantum Algorithms for Sampling Log-Concave Distributions and Estimating Normalizing Constants"
_NeurIPS.cc/2022/Conference — NeurIPS 2022 Accept_

### Official Review · Reviewer_DQxR · 2022-07-07

**Rating:** 3
**Confidence:** 5
**Soundness:** 2 fair
**Presentation:** 2 fair
**Contribution:** 2 fair

**Summary:**

The main contributions of the paper are two-fold.
The first contribution consists of applying quantum estimators of the gradient of the potential function $\nabla f$ in Langevin-type algorithms that sample from a target distribution $\pi(\cdot) = \exp(-f(\cdot))/Z_f$. The authors use Underdamped Langevin algorithm, Metropolis adjusted Langevin algorithm, ULA with randomized midpoint discretization method.
The second part of the paper aims at the estimation of the normalizing constant $Z_f$ using what they call _quantum speedup for MLMC_.
Essentially, this paper is (one of) the first to use the quantum estimation techniques in the problem of MCMC sampling.

**Questions:**

The paper is very hard to read. There no many proofs and the claims are hard to check, as most lemmas and propositions are taken from other papers. The main algorithms should be presented in detail and proofs of the main claims should be stated clearly.

**Limitations:**

The application of this work is not possible at this moment and it is not clear when it will be possible to implement such an algorithm. While as the existing work on Langevin sampling is simple to implement when compared to the proposed methods.

**Strengths And Weaknesses:**

STRENGTHS
The main advantage
of this approach is the low query complexity for each iteration which implements a known zeroth-order gradient estimation algorithm.

WEAKNESSES
The weakness of the paper is the quality of presentation and the applicability of the algorithm. The paper is hard to read, as most notions are presented very briefly. The quantum computations and the estimation algorithm are not well written and are not well defined. The main sampling algorithms are not presented either. This is very inconvenient for an unexperienced reader in any of these two topics.

---

> ### Author Response · Authors · 2022-08-01
> **Official Reply**
>
> We thank the reviewer for their evaluation of our work., However, we strongly disagree with the characterization of our presentation of algorithms and of missing proofs of theorems/lemmas. We included detailed pseudocode for our sampling algorithms and our partition function estimation algorithms. Because we have many results for different conditions using different techniques, some of the pseudocode appears in the appendices. In addition, we have rigorously proved our results. The theorems and lemmas in the main text are either summarizing theorems (Theorems 1.1 and 1.2 are summaries of  theorems in subsequent sections) or informal versions of theorems/lemmas whose main version is presented and proved in the appendices. In the appendices, we have 33 theorems and lemmas, of which 19 are directly cited from other papers as technical tools. The remaining theorems and lemmas are all proved. (Note that Lemma C.6 is implicitly proved on pages 24-26.) Because of the broad scope of our results and the space limit, we had to postpone the formal versions of theorems and their proofs to the appendices to concisely convey our main results in the main text.
>
> While we agree that our work cannot be applied using currently available hardware, we see it as foundational work that advances the theoretical understanding of the power of quantum computers and will be applicable when fault-tolerant quantum computers are eventually built. Considerable research on quantum computing has been carried out following the discovery of Shor’s factorization algorithm, even though its practical application to cryptanalysis may be decades away. In past machine learning conferences, there have been many papers studying quantum speedup for machine learning problems from a theoretical perspective, including the following:
>
> - ICML'21: Quantum algorithms for reinforcement learning with a generative model: Wang, Sundaram, Kothari, Kapoor, and Roetteler
> - AAAI'21: Quantum exploration algorithms for multi-armed bandits: Wang, You, Li, and Childs
> - ICML’20: Quantum Expectation-Maximization for Gaussian mixture models: Luongo, Kerenidis, and Prakash
> - ICML’20: Quantum Boosting: Arunachalam and Maity
> - NeurIPS’19: q-means: A quantum algorithm for unsupervised machine learning: Kerenidis, Landman, Luongo, and Prakash
> - ICML’19: Sublinear quantum algorithms for training linear and kernel-based classifiers: Li, Chakrabarti, and Wu
> - NeurIPS’18: Online Learning of Quantum States: Aaronson, Chen, Hazan, Kale, and Nayak
>
> Therefore, we believe that our work is of general interest to the NeurIPS community even at this moment.

---

### Official Review · Reviewer_rirc · 2022-07-10

**Rating:** 5
**Confidence:** 4
**Soundness:** 3 good
**Presentation:** 3 good
**Contribution:** 2 fair

**Summary:**

This paper discusses quantum type algorithms in estimating the log concave probability measures. The rates in total variation and W2 norm are provided.

**Questions:**

The paper is well written and I do not have question.

**Strengths And Weaknesses:**

The paper is well presented and I mostly enjoyed reading the paper. The results in the paper are scientifically correct and make sense to me.
However, the tools used in the paper to prove the convergence rate are quite standard. The novelty here is to put all analysis in the quantum lens. Moreover, as we can see in Table 1, the quantum versions of ULA or MALA do not really differ from the standard ULA and MALA, regarding the dependence in dimension d. This raised question on whether the quantum setting is really beneficial.

---

> ### Author Response · Authors · 2022-08-01
> **Official Reply**
>
> We are glad to hear that the reviewer found our paper well presented. We would like to better highlight our technical contributions here.
>
> 1. Regarding our quantum ULA algorithms (ULD and ULD-RMM), we are able to use only quantum evaluation (zeroth-order) queries, whereas classical ULA algorithms need to use gradient (first-order) queries. In terms of the dependence on the dimension d, classical algorithms with zeroth-order queries incur another factor of d. Technically, this is achieved in Lemma C.1, where on quantum computers we can compute the gradient of smooth functions using O(1) quantum evaluation queries. This is a fundamental difference and is a benefit not shared by the classical counterparts.
>
> 2. Quantum simulated annealing. We are the first to use the quantum simulated annealing and quantum mean estimation techniques to speed up Langevin dynamics. Specifically, in the normalization constant estimation problem, we achieve a quadratic speedup in epsilon (the accuracy parameter) for all three types of Langevin dynamics.
>
> 3. Effective spectral gap. In analyzing the mixing time of quantum walks, we generalize the notion of the “warmness’’ from the classical Markov chain literature to the quantum setting and show that it has a natural correspondence to the effective spectral gap of the underlying Markov chain, which further characterizes the quantum mixing time. Furthermore, our paper is the first to prove that for a Markov chain with a “warm start’’, the classical mixing time can still be quadratically sped up by the quantum walk.  Prior to our work, quantum advantage was only known for the non-warm case.

---

### Official Review · Reviewer_gFQg · 2022-07-12

**Rating:** 6
**Confidence:** 3
**Soundness:** 3 good
**Presentation:** 3 good
**Contribution:** 4 excellent

**Summary:**

The authors consider the problem of sampling and normalizing constant estimation for log-concave distributions, and derive query complexity bounds for quantum algorithms. The authors use state-of-the-art bounds for classical algorithms as a starting point, and show how to achieve quantum speedup.

They show that for strongly log-concave distributions with condition number $\kappa$:

1. Sampling: Using underdamped Langevin diffusion, quantum algorithms can obtain the same query complexity as classical algorithms while using zeroth rather than first-order (gradient) queries. Moreover, quantum MALA has mixing time that is the square root of classical MALA, $\tilde O(\kappa^{1/2} d)$ and $\tilde O(\kappa^{1/2}d^{1/4})$ from a cold or warm start, respectively.
2. Normalizing constant estimation: Quantum algorithms can speed up normalization constant estimation, obtaining query complexity $\tilde O(\kappa^{1/2}d^{3/2}/\epsilon)$ (with MALA) or $\tilde O((\kappa^{7/6}d^{7/6} + \kappa d^{4/3})/\epsilon)$ (using ULD-RMM), which improves the classical dependence on $\epsilon^2$).

They also show a $1/\epsilon^{1-o(1)}$ quantum lower bound for normalizing constant estimateion, giving near-optimality in $\epsilon$.

Key quantum tools include:

* Jordan's algorithm which gives an estimate of the gradient using O(1) queries to a quantum evaluation oracle.
* A general square-root speedup for quantum algorithms based on reversible Markov chains, when using a sequence of Markov chains with *slowly varying* stationary distributions, starting from the initial distribution. (This is applied to MALA.)
* Quantum algorithm for mean estimation, and quantum version of multilevel Monte Carlo whose query complexity has dependence $O(1/\epsilon)$.

**Questions:**

* How are quantum states over $\mathbb R^d$ defined; what "space" do they live in? It seems that they have to be "square roots" of probability distributions over $\mathbb R^d$. Some care (and measure theory...) is required to make this mathematically formal, but the preliminaries section only discusses the case of finite-dimensional spaces.
* In what sense can one expect to prepare these states (in e.g., a discrete quantum circuit, or another reasonable model of a quantum computer)?

Minor notes:

* line 51: delete "log-concave" - that is a description of the distribution, not the problem. Add: random variable "with distribution" $\tilde \rho$.
* (A.3): missing - sign
* line 527: Besides... also have -> In addition... also has.
* line 637: There seems to be a missing factor of 2: $||x||^2/(2\sigma_i^2)$
* line 643: Missing $\sqrt{\cdot}$, $\ge$ should be $\le$.
* 666: speedup -> sped up
* 704: oracle "satisfying" -> satisfies
* Algorithm 6-7: It seems that the computation of $\tilde g$ can be put at the beginning of the for loop to avoid writing it twice.
* Algorithm 8, line 6: missing tilde.
* 790: register -> registers
* 797: What does "uncompute" mean?
* 849: What is T in the following equation?
* 855: $O(\epsilon)$ should be $O(\beta \epsilon)$.
* 867: Given a state... -> Let ... be a state
* 929: "speedup" the query complexity -> reduce
* 974: "much" complicated -> more

**Limitations:**

The authors acknowledge some limitiations which they present as open questions. Most significantly, do the quantum algorithms based on underdamped Langevin have a better dependence on $d$ and $\kappa$? (Known speedups don't apply to irreversible chains.)

**Strengths And Weaknesses:**

This paper initiates work on the very natural idea of using quantum algorithms to speed up gradient-based MCMC algorithms, establishing basic quantum analogues of the well-studied Langevin-based algorithms and obtaining the signature "square-root" speedups in certain cases. This paves a good foundation for a promising area of research. The observation that quantum algorithms can make do with 0th order information is useful and surprising (for me).

Some of the math regarding the quantum "objects" involved can be made more mathematically precise, especially for those who are not as familiar. I suggest adding more exposition on the quantum formalism.

---

> ### Author Response · Authors · 2022-08-01
> **Official Reply**
>
> We thank the reviewer for their detailed reading and valuable comments, and we are glad the reviewer finds our paper to provide a good foundation for a promising area of research. We will adopt the detailed suggestions in future versions of our paper, and will also add more exposition on the quantum formalism to make the mathematics of our paper more accessible.
>
> Regarding the questions about states in a continuous space R^d and their efficient preparation, we have a discussion in Appendix C.2.2, where we use the techniques of Ref. [6] to efficiently deal with such states. The intuition is that in a realistic implementation, everything is discretized. Similarly to implementations of continuous-space classical algorithms, we use a finite-precision representation of real numbers, while in the analysis, we assume these numbers to be in a continuous space.
>
> We find the question about quantum speedups of underdamped Langevin algorithms in kappa and d to be natural and interesting. However, as we mentioned in our discussion of open questions, the main difficulty is that ULD and ULD-RMM are irreversible, while most available quantum walk techniques only apply to reversible Markov chains. We hope this can be addressed in future work.

---

### Meta-Review · Area_Chair_3bqb · 2022-08-30

**Recommendation:** Accept
**Confidence:** Less certain

**Metareview:**

This work considers the problem of sampling and normalizing constant estimation for log-concave distributions in the quantum setting.
Starting from state-of-the-art bounds for classical algorithms, the authors show how to achieve quantum speedup in a number of settings.
A quantum lower bound for normalizing constant estimation is also derived (as a function of the desired accuracy $\epsilon$). This submission initiates a natural direction --- using quantum algorithms to speed up gradient-based MCMC algorithms --- and obtains interesting and non-trivial improvements in some base cases. The expert reviewers who read the paper in depth found this work conceptually interesting, well-written, and technically non-trivial. Consequently, I recommend acceptance.

**Award:**

No

---

### Decision · Program_Chairs · 2022-09-14

Accept